# Ultra-compact broadband terahertz spectroscopy sensor enabled by resonant-gradient metasurface

Ride Wang[1,9] ✉, Dongze Zhang[2,9], Lu Chen[3,9], Nan Zhang[4,9], Dongxiao Li[5], Rundong Jiang [1], Xiaobao Zhang[1], Xiao Yang[1], Liuyang Zhang [4], Shuming Wang [6] ✉, Xiaogang Liu [7] ✉, Chao Chang [1,2] ✉ & Din Ping Tsai [8] ✉

Research on two-dimensional designer optical structures, especially ultra-thin optical elements dubbed 'metasurfaces', focused on engineering resonances to enrich the broadband spectral information, catering to the increasing sensing and detection demands. Current designs are constrained by discrete sampling of the spectrum, limiting continuous space-to-spectrum mapping. Here we present an integrated terahertz (THz) plasmonic gradient micro-photonic approach for encoding broadband spectra to observe molecular fingerprints. This innovation utilizes smooth variations in the metasurface's unit cells to achieve an extraordinary density of resonances, exciting unique optical modes referred to as bound states in the continuum (BICs) in a non-periodic structure. The device covers a spectral width of 0.9 THz with only 19 modes in a compact $400 \times 125\ \mu m^2$ area. We demonstrate real-time, label-free identification of multiple analytes with enhanced vibrational fingerprints without complex scanning, offering a compact size, broad capabilities, and adjustable resolution to advance portable THz spectroscopy.

The enhanced absorption spectroscopy (EAS) has emerged as a robust technique for non-invasively detecting trace molecular characteristic absorption features with exceptional specificity and sensitivity[1–4]. EAS leverages the localized resonance and enhancement of electromagnetic fields provided by resonant micro/nanostructures, targeting the terahertz (THz) spectra that host molecular vibrational modes. Various collective vibrational and rotational frequencies of fundamental components of life, including nucleic acids, amino acids, lipids and carbohydrates, exhibit distinct spectral absorption features in the THz range[5–7]. The resulting enhanced light-matter interaction facilitates trace amount analyte detection[8,9], with unique molecular absorption features enabling the extraction of specific chemical information and crystal structures from spectrally resolved measurements. Moreover, for general biological processes in mixed systems that involve multiple elements, distinguishing and monitoring individual components of such heterogeneous mixtures is a central goal of biosensing[10,11]. Therefore, the capability to achieve label-free sensing platforms that can independently track the temporal evolution of

[1]Innovation Laboratory of Terahertz Biophysics National Innovation Institute of Defense Technology, Beijing, PR China. [2]School of Physics, Peking University, Beijing, PR China. [3]Key Laboratory of Weak-Light Nonlinear Photonics, Ministry of Education, School of Physics, Nankai University, Tianjin, PR China. [4]School of Mechanical Engineering, Xi'an Jiaotong University, Xi'an, Shaanxi, PR China. [5]Key Laboratory of Optoelectronic Technology & Systems of Ministry of Education, International R & D center of Micro-nano Systems and New Materials Technology, Chongqing University, Chongqing, PR China. [6]National Laboratory of Solid-State Microstructures, School of Physics, Nanjing University, Nanjing, PR China. [7]Department of Chemistry, National University of Singapore, Singapore, Singapore. [8]Department of Electrical Engineering and State Key Laboratory of Optical Quantum Materials, City University of Hong Kong, Kowloon, Hong Kong, PR China. [9]These authors contributed equally: Ride Wang, Dongze Zhang, Lu Chen, Nan Zhang. ✉e-mail: wangride@mail.nankai.edu.cn; wangshuming@nju.edu.cn; gwyzlzssb@pku.edu.cn; chmlx@nus.edu.sg; dptsai@cityu.edu.hk

absorption fingerprints across a broad spectral bandwidth and distinguish between different biomolecular species holds great potential in unveiling the complex interactions of analytes in biological systems and beyond.

Recently, in order to accurately detect the multi-fingerprints spectrum of the analyte, plasmonic or dielectric micro/nanostructures that support a multi-band spectral response have been designed to achieve this goal[11–15]. Implementation of multi-resonant plasmonic metasurfaces has been used to probe multiple distinct wavelength ranges by exploiting self-similar geometry[11,16], combined with functionalized 2D material[17–19], incident angle of light[20], or other approaches[21–24]. It should be emphasized that resonances of the units are highly susceptible to environmental perturbations, resulting in the overlap of the resonances and fingerprint misalignment. Then, imaging-based molecular barcoding with pixelated dielectric metasurfaces covering a larger spectrum region is exploited[25]. In addition, the broadband waveguide modes supported by a microrod array metasurface or spoof surface plasmon are studied to recognize biomolecules[26,27]. However, the above solutions face large printing areas, complex fabrication processes and the presence of cut-off frequency challenges, respectively.

Additionally, it is crucial to highlight that within the THz frequency range, THz plasmonic metasurfaces offer advantages in terms of reduced experimental errors and cost-effectiveness compared to dielectric metasurfaces, attributable to the comparatively larger dimensions of the unit structure. When combined with biochemical functionalization[28–30], plasmonic structures further leverage their unique strengths, including strong evanescent fields and high sensitivity to molecular binding events, significantly improving detection capabilities. Through the strategic design and manipulation of plasmonic metaatoms, it becomes feasible to manipulate the interference of multiple resonances on these microstructures, thereby enabling precise control over near-field distributions[31,32]. These approaches present a viable strategy for facilitating more practical and highly sensitive biosensing applications, enabling the development of ultracompact, efficient, and significantly more economical sensing tools.

The concept of bound states in the continuum (BIC)[33,34] has become a powerful tool for augmenting the electric field at the nanoscale, holding great potential for amplifying light-matter interactions. Although optical modes exist within the continuum spectrum of the radiative states, they are spatially segregated from the external surroundings. An ideal BIC demonstrates confinement of the optical field within the system and cannot be excited from the far field owing to the complete disruption of its coupling channel to the external environment[35–37]. However, in practice, controlled radiation loss is essential to harness the benefits of intense field amplification. By introducing a leakage channel by breaking the geometrical symmetry of the system, the BIC can be converted into a quasi-BIC (QBIC) with a finite and high Q-factor[38–40]. The conversion allows practical access to strong localized fields with large resonance Q factors. The management and modulation of radiation loss can be precisely controlled by the degree of asymmetry. This method holds vast potential for enhancing light-matter interaction. Several attempts have been made to utilize QBIC for improving biosensor performances, but most of the reported QBIC-based sensors operate only in a narrow-band or finite bandwidth, preventing the differentiation of different biomolecules of a mixture or multi-component compact devices.

Here, we leverage the inherent flexibility of THz plasmonic metasurfaces and introduce an innovative unified gradient metal metasensor toolkit, capable of facilitating in situ ultra-wideband high-sensitivity sensing and bio-specific substance identification (Fig. 1). Our innovative design is crafted by assembling arrays of metaatom while gradually tuning them along one direction in its physical space, forming a localized super-metaatom with high-density spatial resonance manipulation. The gradient variation rate of the metaatom dimensions is carefully optimized to ensure the required resonator size for a compact device footprint. By exploiting the continuous spatial variation of resonant frequency in gradient metasurfaces, we demonstrate real-time extraction of THz absorption fingerprint spectra of specific molecules in a complex scenario bioanalytical hybrid system. We perform trace detection of biomolecules, including neurotransmitters and their metabolic intermediates with multiple absorption bands in an ultra-wideband operating range, confirming the excellent capability of our method for mixture identification and component detection. We then use a machine learning approach to

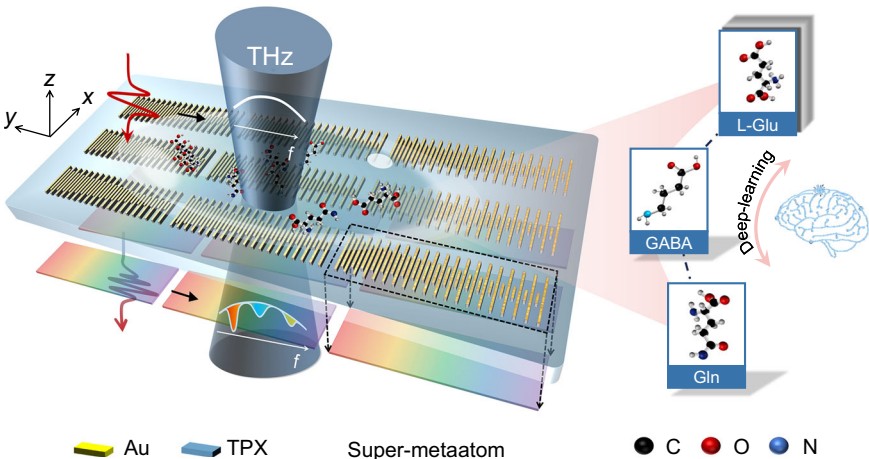

**Fig. 1 | Concept of the THz plasmonic resonance-gradient metasurface.** Schematic diagram of the gradient metasurface formed by introducing variation of the geometrical parameters along the *y*-axes. The super-metaatom is an extension of the metaatom concept, constituting a composite unit structure composed of metaatoms with different functionalities (e.g., variable size). The electric field of the broadband excitation source is aligned with the varying long axis (*x*) of the microbars. The resonance frequencies are tuned over a continuous spectral range. Resonance frequency coverage enables continuous spectral mapping. Simultaneous enhancement and detection of changes in analyte absorption spectra are achieved by arbitrarily designing gradient microbars to promote broadband enhancement. The design enables overlap with the vibrations of the characteristic absorption bands of the L-glutamic acid(L-Glu), γ-aminobutyric acid (GABA) and glutamine (Gln). The platform works stably, allowing for real-time detection of dynamic changes in characteristic fingerprints and quantitative identification with the help of machine learning in multi-analyte systems. The machine learning process extracts multi-dimensional sensing information to identify molecules and distinguish the proportions of mixtures.

determine the relative concentrations of the three molecule mixtures to overcome the challenge of quantitatively differing component absorption intensities. In addition, specific gradient metasurfaces designed using this method can be generalized to a wider spectral range to match the absorption fingerprints of the target molecules. The tailored sampling density is designed to minimize the footprint of the metasensors, showing an order-of-magnitude reduction over the pixelated metasurface design[25]. From an application perspective, gradient plasmonic metasurface fingerprint detection devices not only help to reduce manufacturing costs, but also allow for robust in situ broadband measurements for trace samples, such as biomolecules. Through comprehensive analysis of the spectral and coupling parameter space, this method reveals a generalized broadband EAS (enhanced absorption spectroscopy) technique via metasurface engineering, significantly expanding its application potential to include on-chip spectroscopy and real-time dynamic reaction monitoring.

## Results

### Gradient metasurface for multi-QBIC broadband resonances

We employ a distinctive resonant design based on microbar arrays supporting multiple QBIC resonances that facilitate the simultaneous detection of a wide range of molecular vibration fingerprints at multiple spectral points through robust near-field interactions. As shown in Fig. 2a, the THz plasmonic gradient metasurface consists of microbars (referred to as metaatoms below) with varying feature sizes, each supporting a QBIC with a high but finite Q-factor at its own resonant frequency. When excited by an inherent broadband light source with a bandwidth covering all the resonant frequencies involved, all metaatoms are simultaneously excited, as the simulation in the bottom panel of Fig. 2a (detailed parameters of the gradient metaatoms are presented in Table S1). Fig. 2b shows the geometry of one exemplary metaatom made of three gold microbars on the TPX substrate, where the microbars on the sides have the same length $L$, whereas the middle microbar has a length of $L_1$. A key parameter $\delta = (L - L_1) / 2$ is defined to represent the scaling of the gradient metaatoms. Fig. 2c presents simulated transmission spectra for individual metaatom structures of the gradient metaatoms in Fig. 2a with fixed periodic boundaries ($P_x \times P_y = 125\,\mu m \times 40\,\mu m$), where each spectrum corresponds to the response of an isolated unit cell, covering a broadband spectrum ranging from 1.2 THz to 2.1 THz. This design method is highly universal and also applicable to complementary structures. Note that with judiciously designed gradient metaatoms, our approach can offer extended spectral coverage that is tailored for a wide range of user-specific applications. Additionally, we quantitatively evaluated the uniform electric field enhancement of resonances across the entire broadband range in Fig. 2d. As $\delta$ increases, the resonant frequency of the metaatom increases, while the Q-factor and the electric-field intensity of the resonances decrease (Fig. 2e). The relatively low Q-factor in this system is due to scattering and material losses caused by the inevitable rough surface and finite area of the metasurface, which limit the resonance linewidth.

Figure 2f–h compares different designs of the THz plasmonic metaatoms, whose unit cell consists of two subgroups of gold microbars. When the microbars share the same dimensions, as shown in the left panel of Fig. 2f, adjacent three microbars form a pair and collectively exhibit an electric Fano resonance with strong ED and TD excitations, resulting in a symmetry protected BIC mode that is decoupled from the exterior environment, as evident by the lack of resonance peaks in the simulated transmission spectrum (left panel in Fig. 2g)[41]. As the size of the alternative microbars is changed, as shown in the middle panel of Fig. 2f, such symmetry is broken, transforming the perfect BIC mode into a QBIC mode with a high but finite Q-factor shown in the middle panel of Fig. 2g and red-boxed panel in Fig. 2h. Similarly, if the sizes of the alternative microbars are further perturbed to scale gradually, additional QBIC modes with

different resonance frequencies can be supported by such a THz plasmonic gradient metasurface (right panel in Fig. 2g and red-, orange-, green-boxed panels in Fig. 2h). The spacing between resonant frequencies is determined by the difference of the defects. In order to further confirm the performance of the constructed metasurface, numerical simulations utilizing the 3D finite-difference time-domain (FDTD) solution software are performed. We detail the design process of a gradient metasurface supporting broadband-enhanced THz spectroscopy in Fig. S2. The novelty of the approach outlined above stems from the fact that by judiciously designing the scaling of the microbars in adjacent metaatoms, continuous mapping of the resonant spectrum can be achieved with a single THz plasmonic gradient metasurface, offering multiplex advantages including a reasonably small footprint, dramatically enhanced light-matter interaction, full coverage of a broad bandwidth, and high design flexibility customized to comply with various biological applications.

To demonstrate our ability to accurately differentiate molecule-specific information from complex analytes, we first numerically tested, as an illustrative example, a mixture of L-Glu, GABA, and Gln using our THz plasmonic gradient metasurface. These neurotransmitters and their metabolic byproducts play a pivotal role in elucidating the transmission of neural signals, regulating neural system functionality, and understanding the onset, progression, and treatment of related neurological disorders[42,43]. Fig. 3a shows a schematic design of our metasurface device, where alternating microbars are assigned gradually decreasing sizes to cover the absorption bands of L-Glu, GABA, and Gln. The relative spectral efficiency advancement of the gradient metasurface is quantified by $\Delta f/(f_0 \cdot D)$, achieving $10.91\,mm^{-2}$, where $\Delta f$ represents the operational bandwidth, $f_0$ denotes the central working frequency, and D corresponds to the physical area of the metasurface required to achieve broadband functionality. The detailed comparative analyses are outlined in Table S2 and Fig. 3b. The measured barcode spectrum of the mixture is shown in Fig. 3c. Simultaneous amplification and detection of small changes in analyte absorption spectra are achieved by carefully designing a sequence of THz microbars to promote multi-band enhancement at desired resonance positions. Here, the THz plasmonic gradient metasurface consists of 19 groups of microbars, each supporting a resonant peak ranging from 1.2 to 2.1 THz. The high number of resonant peaks, together with their finite peak width, results in a continuous spectral coverage, as shown in the measured spectra in Fig. 3d, e. We then tested the gradient metasurface design with a simulated mixture of analytes in Fig. 3f, g. Compared to the bare metasurface, spectral intensity modulations that closely match the characteristic absorption bands of L-Glu, GABA and Gln are observed in the presence of the analyte. This THz plasmonic-gradient-metasurface-enabled platform offers simultaneous detection of multiple biological samples, and can also be extended to perform dynamic investigation of complex analytes.

### Gradient metasurface enables multi-analyte detection

Next, we further experimentally demonstrated the capability of the THz plasmonic gradient metasurface for real-time, in-situ, and dynamic tracking of multiple biological analytes, showcasing its potential in probing the identification of complex molecules as well as analyzing dynamic changes in concentration. The metasurface chip is integrated with a microfluidic chamber where analyte solutions with different mixtures and concentrations interact with the metasurface as shown in Fig. 4a. Broadband (0.5–3.5 THz) THz radiation with normal incidence is transmitted through the solution and gold microbars from above, and the remaining THz radiation is collected by a spectrometer. The transmission spectra of the bare metasurface and metasurface in the solvent were compared in Fig. 4b. Note that by varying the tuning size and the variation rate of the gradient metasurface design, the

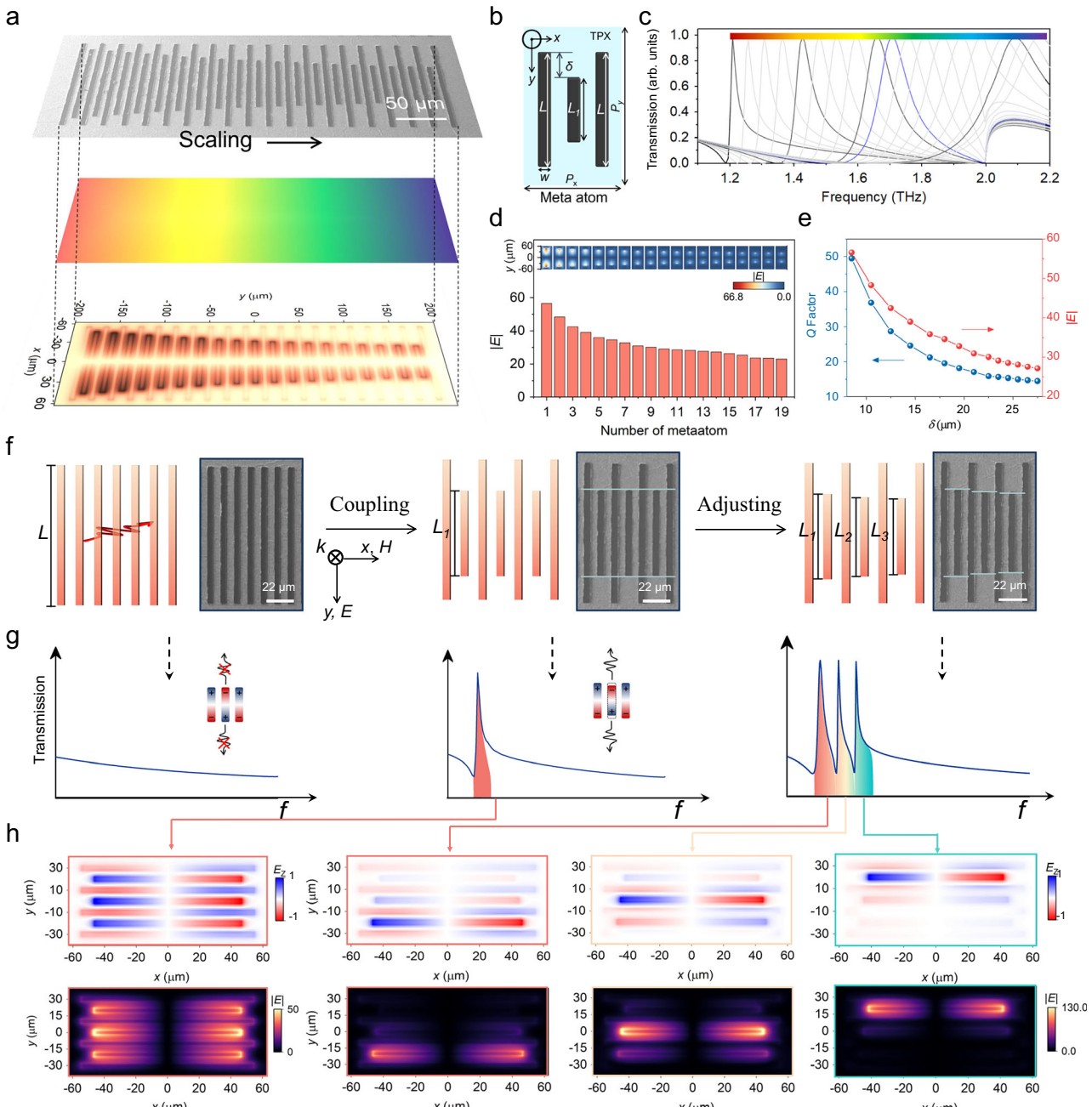

**Fig. 2 | Design of broadband spectral-based THz plasmonic gradient metasurface. a** Broadband performance on TPX patterned with THz plasmonic gradient metasurfaces. Scanning electron microscopy (SEM) image (Fig. S1) of a spectral gradient with the scaling along the $y$-axis. The bottom panel shows that all the varying metaatom are excited simultaneously by the THz time domain pulse signal. The parameters of the gradient microrods are shown in detail in Table S1. **b** Schematic of the metasurface unit cell, which comprises a group of gold microbars on TPX substrate ($w = 5\,\mu m$, $L = 110\,\mu m$, $P_x = 125\,\mu m$, $P_y = 40\,\mu m$). $L_1$ is the dimension of the length-varying microbar. **c** Each normalized simulated transmission spectrum was taken from the independent periodic metaatom spectral gradient along the $y$-axis, as shown in gray. The colored spectra correspond to the mono-spectral metasurfaces, each normalized to the gradient's maximum transmission at its respective spectral position. Resonance is highlighted with a blue line. **d** Quantitative assessment of resonance uniformity enhancement over the entire broadband range. **e** $Q$-factor and electric field intensity of the QBIC resonance plotted against different scaling increments for gradients with $\delta$. **f** Schematic illustration of continuous resonant metasurface whose unit cell is made of two subgroups of gold microbars. The lengths of all microbars are intentionally perturbed to create multi-QBICs. The right part of each schematic is the corresponding SEM image. **g** The corresponding transmission spectrum of the metasurfaces in (**f**). The insets show the mechanism of the interfering resonances for a resonating system. **h** Calculated maps of electric field enhancement in the metasurface unit cell for linearly polarized excitation at the resonant wavelengths of different QBIC modes ($E$-field along the $x$-axis).

shape of the transmission spectrum can be engineered for various user-specific applications (see two representative examples in Fig. S4 and Table S3). Vertical black and orange lines label the start/end positions of the broad-spectrum response of the gradient metasurface in the presence or absence of edible oil solvent (Refractive index $n_{oil} = 1.47$). A reduction in transmission is observed due to the oil's inherent THz absorption, but no solvent-specific peaks interfered with the target biomolecular fingerprints. Fig. 4c shows the characteristic

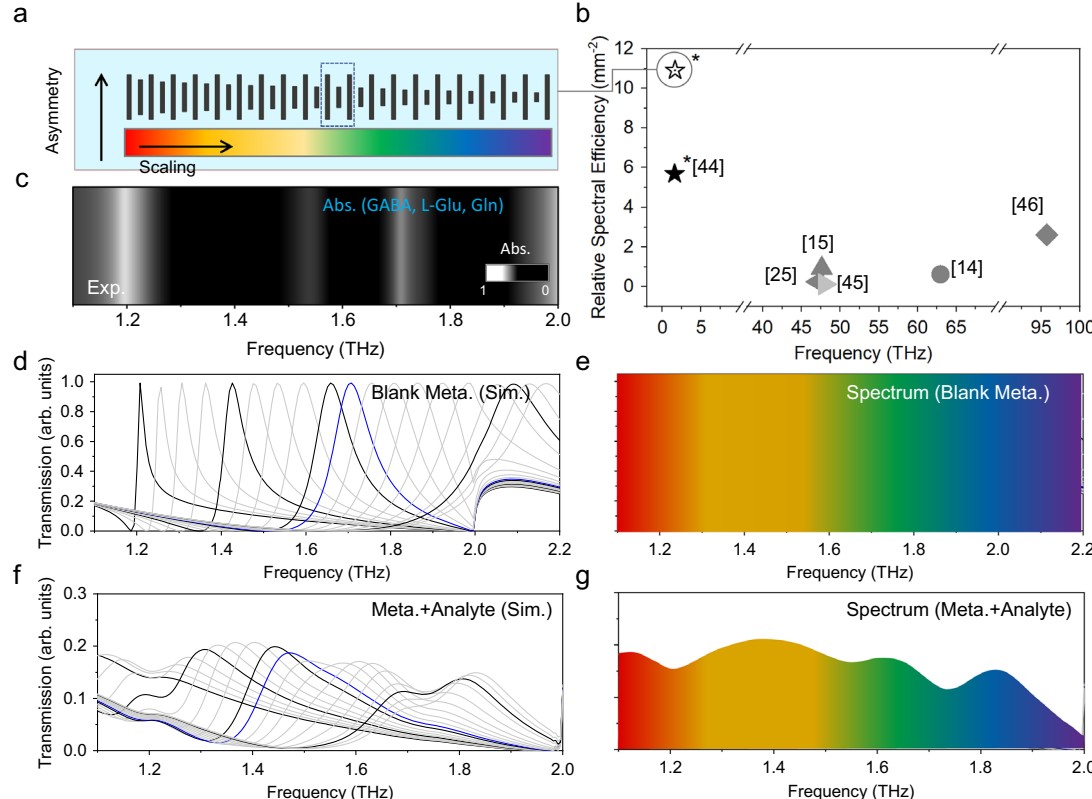

**Fig. 3 | Differentiating molecule-specific information from complex analytes.**
**a** Graphical representation of a gradient metasurface. The in-plane dimensions of the super-metaatom decrease gradually from left to right. **b** A plot of relative spectral efficiency versus the operating frequency. Comparative works are marked with numbers: dual-gradient metasurfaces (ref. 15), pixelated sensors (refs. 25,44), gradient dielectric metasurfaces (refs. 14,45). Our work is highlighted with an asterisk (*), showcasing the multiband (ref. 46) and the gradient metasurface, respectively. In addition, we summarize and compare the literature in the Table S2. **c** Spatial barcode of the mixture (L-Glu, GABA and Gln) absorption extracted from

the measured data. **d, e** Normalized simulated transmission spectra of a bare metasurface (color-encoded) with varying $\delta$ illustrating the continuous spectral coverage with resonant peaks. Each spectral line is obtained from a metasurface composed of periodically arranged microbars of the same size. The resonance highlighted in blue corresponds to the area marked with the same color in (**a**). The operating range of the device is about from 1.2 to 2.1 THz. **f, g** Normalized simulated transmission spectra of a metasurface covered with mixture analyte. Intensity modulations are observed at the characteristic absorption frequencies of the mixture.

absorption barcodes of Gln, GABA, L-Glu, together with their mixture, as a reference for identification for the following experiments.

**Real-time dynamic sensitivity monitoring**

To demonstrate the dynamic monitoring performance of our THz plasmonic gradient metasurface, a set of continuous time-domain measurements was performed, as shown in Fig. 4d, where various concentrations of L-Glu solutions were monitored every 3 s. The electric field amplitude fluctuation $\Delta E_{\text{max-min}}$ and its stability over the 612 s of continuous monitoring are plotted in Fig. 4e. $\Delta E_{\text{max-min}}$ accurately reflected changes in analyte mass (0 μg to 16 μg), as evidenced by repeatable and quantifiable $\Delta E_{\text{max-min}}$ levels. Further, to rigorously validate quantitative detection capability, we introduced a "blind" test (gray ribbon area in Fig. 4d), where Researcher A controlled 12 μg of analyte in the microfluid chamber, and Researcher B, without prior knowledge of the analyte mass, performed the measurements. From the experimental data, Researcher B then derived the expected analyte mass, which matches the designed value of 12 μg. This test confirms the system's ability to successfully differentiate biomolecule concentrations without prior knowledge. The transmission spectra shown in Fig. 4f were obtained through Fast Fourier Transform (FFT) the time-domain signals in Fig. 4d. The transmission spectra clearly revealed distinct concentration-specific L-Glu absorption features around 1.22 THz. Residual analysis, as shown in Fig. 4g, validated system

reliability, showing minimal deviations from the zero-reference line and consistently high signal-to-noise ratio (SNR ≈ 28 dB) throughout the 612 s measurement. These results collectively demonstrate the robustness, precision, and long-term stability of our dynamic monitoring approach.

Our THz plasmonic gradient metasurface not only preserves the intrinsic spectral features of the analyte (as evidenced by Fig. 4h, i), but also enhances the detection sensitivity through metasurface integration, making our system an ideal platform for identifying even more complex solutions with multiple mixed analytes. In Fig. 4h, the time-domain signal and the corresponding transmission spectra of different solutions with various mixtures, all measured without the presence of the metasurface, are plotted. In comparison, Fig. 4i shows a set of measurements obtained via the THz plasmonic gradient metasurface incorporated system. As shown in Fig. 4i, we start by adding 28 μg of GABA to the solvent (black curve) and gradually add L-Glu from 6 μg to 22 μg (pink curves), while monitoring the transmission spectrum. As the concentration of L-Glu increases, the absorption dip around 1.22 THz becomes more and more prominent, which matches very well with the expected characteristic absorption band of L-Glu. In addition, 10 μg, 14 μg, and 18 μg of Gln are subsequently mixed with a GABA (28 μg) and L-Glu (18 μg) mixture, respectively, and their transmission spectra are shown in the blue curves. Once Gln is introduced to the mixed solution, an absorption dip around 1.75 THz is observed and the

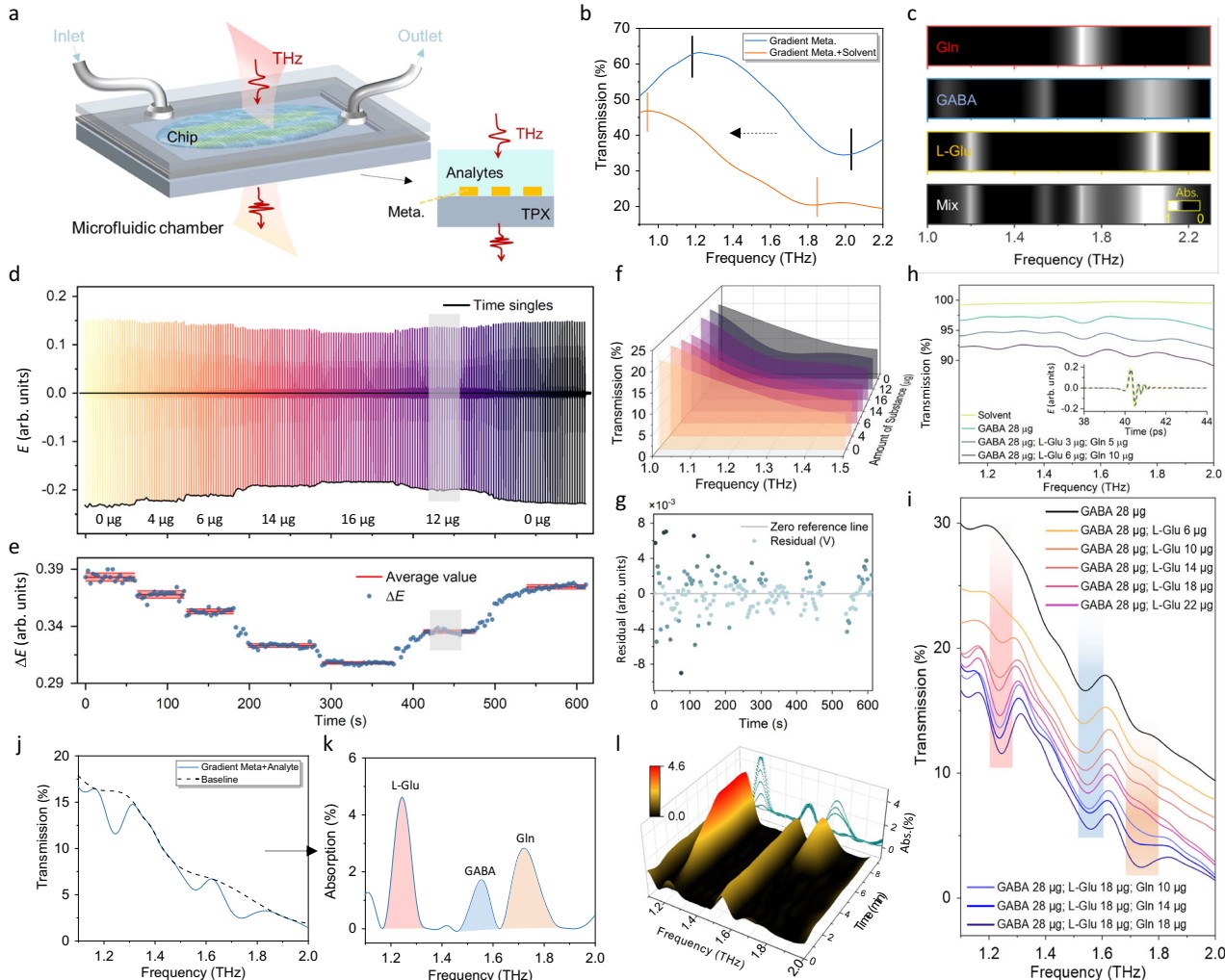

**Fig. 4 | Simultaneous monitoring of multiple biological analytes. a** Schematic of the chip integrated with a microfluidic chamber (height: 1 mm) for analyte detection. The experimental setup is shown in Fig. S3. **b** Transmission spectra of the gradient metasurface before/after edible oil coating (blue/orange solid lines), covering a wide spectral range of 1.2 to 2.1 THz. **c** Measured absorption barcodes for three species pure/mixed biomolecules. **d** Time-domain electric field amplitude, collected at 3 s intervals, change dynamically with varying L-Glu masses (0, 4, 6, 14, 16, 12, and 0 µg). The gray band predicts the mass for unknown samples. **e** The average electric field amplitude (center solid red line) over 612 s of continuous monitoring. The amplitude fluctuations ($\Delta E_{max-min}$) reflect mass sensitivity. The red shaded area represents the standard deviation. **f** Transmission spectra obtained by fast Fourier transform (FFT) of the time-domain signals show the specificity of L-Glu

characterized by its absorption at 1.22 THz. **g** Residual analysis confirms minimal deviation from the zero-reference line. **h** Reference transmission spectra and time-domain signals of analytes without metasurface enhancement are provided. **i** Spectra for L-Glu mass varying from 6 µg to 22 µg (pink solid line), with GABA fixed at 28 µg (black solid line). The blue series shows spectra obtained by adding Gln, with mass varying from 10 µg to 18 µg, to a mixture of fixed L-Glu and GABA masses. The red, blue, and orange rectangular positions indicate the characteristic absorption spectra of L-Glu, GABA, and Gln, respectively. **j** Transmission spectra (18 µg Gln mixed with a mixture of GABA (28 µg) and L-Glu (18 µg)) and dashed baseline-corrected signals are shown. **k** Absorption spectrum calculated from the transmission spectra in (**j**). **l** Real-time 3D plots of differential absorbance versus frequency and time (1-min intervals).

dip depth increases with increasing Gln density, also as expected. The steep roll-off phenomenon observed in the broad spectral range is caused by the combined effects of different radiation losses from metaatom with varying sizes and sample-induced Rayleigh scattering, neither of which affects the interpretation of our core findings. The results are obtained by performing an FFT on the time-domain signal window (34–46 ps), where stable time-domain signals are continuously collected under different conditions, with the results displayed in Fig. S5. Further analyses of the transmission peaks are detailed in Fig. 4j, k. The dashed curve indicates the calibrated baseline, which was obtained using an asymmetric least square smoothing (AsLSS) fitting algorithm[17]. By subtracting the baseline from the measured spectrum, we can extract the absorption features of the sample.

Then, we quantitatively evaluate the enhancement factor (EF) of the enhanced THz absorption spectroscopy (ETHzAS), which is defined as[12]:

$$EF = \frac{I_{ETHzAS} - I_{ref}}{I_{ref}} \times \frac{N_{ref}}{N_{ETHzAS}} \quad (1)$$

where $I_{ETHzAS}$ and $I_{ref}$ (Fig. S6a) represent the absorption intensity measured with and without the metasurface, respectively. The term $N_{ref}$ and $N_{ETHzAS}$ correspond to the number of molecules contributing to absorption in the reference case and within the metasurface's localized field-enhanced region, respectively. It can be obtained by calculation that the EF is found to be about 80 (Fig. S6b). Finally, we

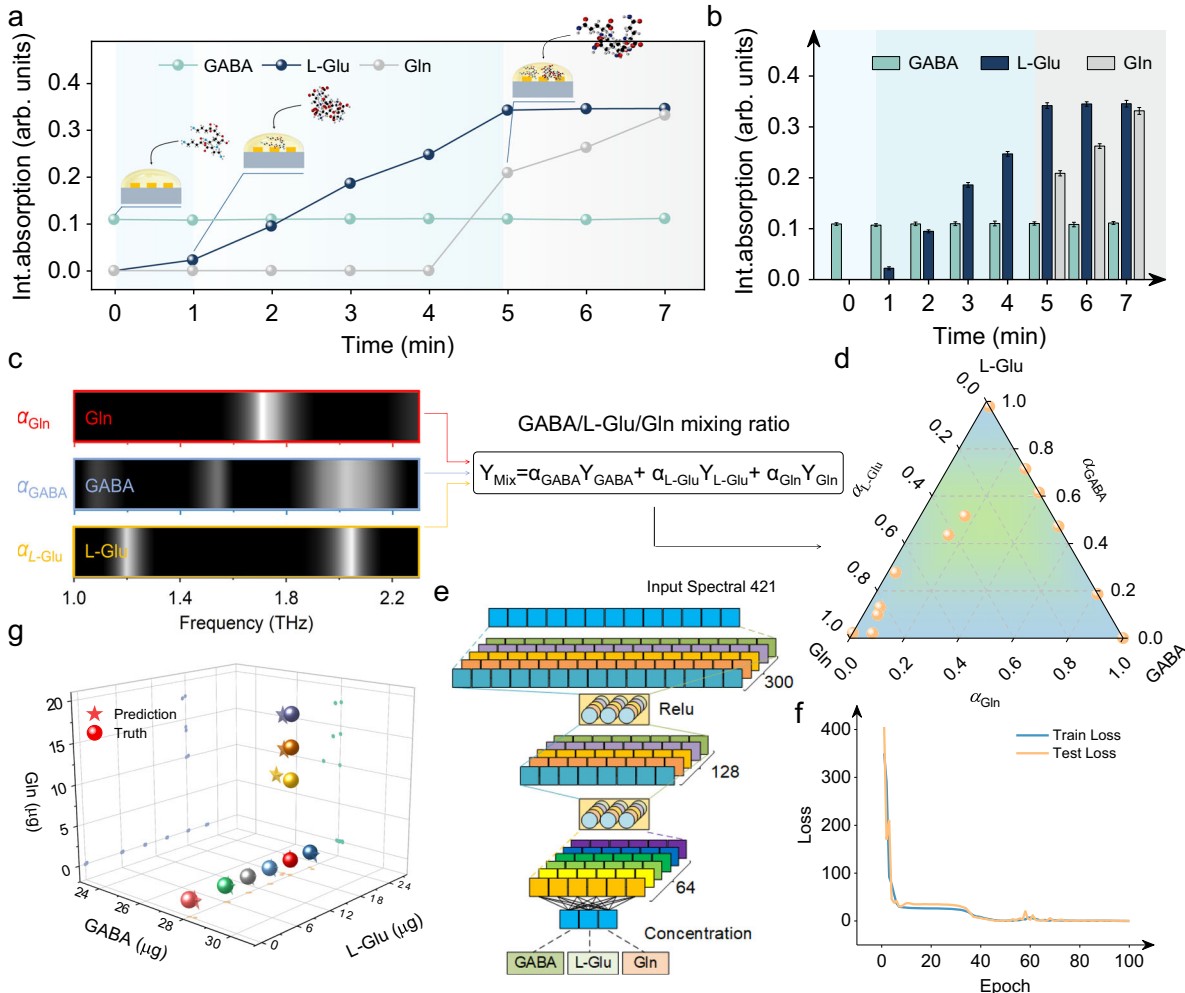

**Fig. 5 | Multimodal analysis and concentration prediction using deep neural networks (DNN). a**, **b** Time traces of the integrated absorbance signal in the L-Glu (red-shaded area), GABA(blue-shaded area), and Gln (orange-shaded area) bands. **c** Molecular barcodes of L-Glu, GABA and Gln reveal the distinct absorption fingerprints. Mixed-state $Y_{Mix}$ with multiple mixing ratios was analyzed. **d** The percentage of the three substances in the mixed analyte. The position of the solid orb on the ternary graph represents the ground truth ($\alpha$, $\beta$, $\gamma$), as known from the mixing stage. **e** A DNN model with three hidden layers, containing 300, 128, and 64

nodes, respectively, was used to predict the concentration of the three analytes. The model was configured with the following hyperparameters: a learning rate of 0.001, 100 epochs, and an early stopping strategy. The activation function used was ReLU. The experimental data were divided into training and test sets in a ratio of 8:2. The training sets were used to train the model, and the loss during the training process is shown in (**f**). After training was completed, the final model was evaluated using the test set. The evaluation results are shown in (**g**).

derive an approximate unified limit of detection (LOD) for the analyte of approximately 10 μg/mm² (mass per unit area).

The dynamic evolution of the transmission spectrum under different mixture conditions is summarized in Fig. 4l. We quantitatively analyzed the signal and noise performance and demonstrated the ability of the system to support our sensitivity requirements in Fig. S7. It is evident that this set of experiments validates the applicability of our THz plasmonic gradient metasurface in real-time, in-situ, dynamic monitoring of complex biological analytes. We'd like to note that this approach is efficient, robust, and can be applied to enable the detection and distinction of a wide variety of biomarkers, amino acids, proteins, and other types of biological samples provided their characteristic absorption bands are in the THz range.

**Concentration prediction of mixed analytes via DNN**

In situ sensing applications frequently encounter the challenge of distinguishing trace amounts of multiple analytes within complex samples. We demonstrate that a THz plasmonic gradient metasurface sensor, by integrating its broadband sensing capability with machine learning methods, can simultaneously monitor and differentiate multiple bioanalytes even with a minimal amount. Initially, we mix three polymers-L-Glu, GABA, and Gln-in varying volumes to create solutions with different compositions ($\alpha$, $\beta$, $\gamma$), as illustrated in Fig. 5a, b. These solutions are then spin-coated onto the gradient metasurface, and their respective absorption barcodes are recorded. The barcodes of pure L-Glu, GABA, and Gln layers, as shown in Fig. 5c, provide the foundation for spectral decomposition. The analysis of the polymer mixtures is presented in the ternary mixture plot in Fig. 5d. Principal component analysis (PCA) is performed on the spectral data corresponding to all mixture ratios. The principal component loadings are depicted in Fig. S8, with each cluster representing a distinct mixture ratio. The lack of overlap between clusters allows for the classification of different mixture ratios.

To more comprehensively capture the variations in analyte concentrations and enhance prediction accuracy, we employ a DNN to construct a regression model for predicting the proportions of mixed analytes. As depicted in Fig. 5e, this DNN model comprises an input layer, three hidden layers, and an output layer. Each hidden layer

consists of 300, 128, and 64 nodes, respectively. The output layer comprises three nodes, corresponding to the three substances in the samples. The transmission spectra from Fig. 4i are then fed into the DNN model as a training set. After 100 training epochs, the network loss steadily decreased and eventually stabilized, achieving a final loss of 0.15 and an accuracy of 93.31%. Subsequently, newly measured spectral data were input into the trained DNN model as a test set. Fig. 5g presents the predicted concentrations of all components in the output mixture, with the predictions closely aligning with the actual values. It is important to highlight that our platform is not limited to detecting three biochemical substances. Owing to its general applicability, it can be extended to biosensing assays involving more biological components, as well as exploring unknown samples (Fig. S9) or analytes with unidentified absorption spectra. This molecule-specific and label-free detection method enables precise analysis of absorption fingerprints, thereby opening exciting avenues for studying biological effects in the THz range. Our findings pave the way for the future development of novel, rapid, label-free, and highly sensitive metamaterial biosensors.

## Discussion

In conclusion, we leverage the versatility of THz plasmonic metasurfaces to achieve in situ ultra-broadband THz sensing and biological fingerprint identification with high sensitivity by implementing a unified gradient metasurface toolbox. By judiciously tailoring the scaling of gold microbars in our design, multiband QBIC resonances can be engineered to form a continuous broadband spectral coverage in the THz range. We demonstrate real-time, in-situ, dynamic detection of complex biological analytes using our THz plasmonic gradient metasurface. By combining with DNN analysis, we also demonstrate trace detection and concentration determination and prediction of mixed analytes, overcoming the challenge of quantitatively identifying multiple bio-analytes in a single measurement. Such an approach paves the way towards sensitive and versatile miniaturized robust THz spectroscopy devices. In addition, such a device can be reused after cleaning, making it a cost-effective alternative for practical applications. Our plasmonic gradient metasurface design, though it is demonstrated only in the THz range due to the great biological interest, can be simply generalized to other wavelength regimes, thus opening up new avenues for clinical analysis and beyond.

## Methods

### Numerical simulations

We conducted FDTD numerical simulations using the Lumerical FDTD software to calculate the transmission spectra and confined near-field electric field distributions corresponding to the resonance modes. The structure is excited by normally illuminating it with plane waves aligned with the dipole microbars and by combining all propagating Floquet modes. In free space, periodic boundary conditions are applied along the x- and y-directions, while perfectly matched layer (PML) absorbing boundary conditions are applied along the propagation of the electromagnetic waves (z). Converged results can be achieved by setting the grid size to be smaller than the corresponding minimum structural dimension to compute scattering properties. The trimer structure consists of gold microbars with a fixed width of 5 μm, deposited on a TPX substrate with a thickness of 2 mm. The dimensions of the metaatoms were systematically varied by adjusting the length of the middle bar in the trimer, with lengths ranging from 48 μm to 93 μm. The material of the microbars is modeled as a perfect electric conductor (PEC), with the conductivity of traditional metals in the THz region approximately $10^7$ S/m, effectively suppressing Ohmic losses. The material parameters, representing the mixture of L-Glu, GABA, and Gln, were derived from experimental data, specifically the real and imaginary parts of the refractive index obtained using palletization techniques. We set the thickness of the analyte to 6 μm in order to be

closer to the experimental setup as an example to illustrate the sensing performance of the device. Through literature research, the most reasonable model is to set the analyte to semi-infinite to elaborate the sensing capability of the sensor. The transmission function returns the amount of power transmitted through power monitors, normalized to the source power. Specifically, the transmission $T(f)$ is defined as the ratio of the transmitted power $P_{trans}(f)$ to the incident power $P_{inc}(f)$ at each frequency $f$, expressed by the formula $T(f) = P_{trans}(f) / P_{inc}(f)$.

### THz spectroscopy system measurements

To acquire spectral signals, we used a THz time-domain spectroscopy (THz-TDS) system (Advantest Co., Ltd., Tokyo, Japan; TAS7500TS) with a spectral range of 0.5 – 7 THz, a frequency resolution of 3.8 GHz, and a scanning range of 262 ps. The system employs an asynchronous optical sampling (ASOPS) configuration for time-delay generation, offering key advantages: (i) elimination of mechanical delay stages for faster scanning (262 ps range at a 50 MHz repetition rate), (ii) superior timing stability (<100 fs), which is critical for high signal-to-noise ratio (SNR) measurements, and (iii) direct compatibility with 1550 nm fiber laser sources. The setup consists of two fiber lasers (pulse duration <50 fs, 50 MHz repetition rate), photoconductive antennas serving as both THz transmitter and detector, a precision time-delay device, and data acquisition software. One laser excites the THz signal, while the other detects it, enabling precise time-domain measurements in reflection and transmission modes with a system noise level below 60 dB. For this study, transmission measurements under dry air conditions were used to facilitate experimental control and dynamic monitoring.

### Fabrication of sensor chip microstructures

I) TPX substrate preparation: The substrate was subjected to a combination of injection molding and precise cutting to achieve the desired dimensions. This was done with a polymeric material with a refractive index of approximately 1.46. Then, the substrate was cleaned with ethanol and deionized water and dried with a gentle stream of nitrogen gas. II) Metal Pattern Preparation: i) Lithography: Photoresist was spin-coated onto the substrate and baked to cure. Then, using UV irradiation, the pattern of the mask plate was transferred to the photoresist coating. ii) Development: The irradiated substrate was reacted in the corresponding developer. Noted that the development time should be precisely controlled. Specific wavelengths of light are used in combination with the corresponding developer to ensure efficient exposure and development. iii) Metal deposition: A 10 nm chromium layer and a 100 nm gold layer are deposited on the substrate using a magnetron sputtering system. The purpose of the chromium layer is to enhance the adhesion during the metal deposition process. iv) Stripping process: The coated samples were immersed in acetone to finely remove the photoresist layer and gold film adhering to the top edge. Ultrasonic debonding techniques were used more efficiently to maintain the integrity of the metal microstructure, aiming to prevent any undesired damage. In addition, we repeated the mechanical and structural stability tests of the metasensor before and after solvent exposure, and the results demonstrate that the device maintains excellent broadband enhancement performance (Fig. S10).

### Machine learning

In the classification of mixed samples, we employed PCA to identify the principal components within the measured spectral data, projecting high-dimensional data into a lower-dimensional space while striving to preserve the variance information, thereby facilitating the visualization of the feature space. PCA was performed using Python, with detailed steps including data preprocessing, calculating the covariance matrix, deriving eigenvalues and eigenvectors, ranking them by the magnitude of eigenvalues, selecting the top k principal components, and mapping the data onto these chosen components. For concentration prediction, we developed a multilayer feedforward neural

network based on Python 3.9. Utilizing multiple hidden layers and ReLU activation functions, the network captures complex patterns in the input features through layer-by-layer nonlinear transformations, ultimately outputting the target values. The training and validation datasets comprised 37,890 spectro-temporal data points acquired through time-resolved spectroscopic monitoring of dynamic ternary mixtures. The model architecture consists of an input layer, three hidden layers, and an output layer. The hidden layers contain 300, 128, and 64 neurons, respectively, employing ReLU activation functions. The output layer, comprising three neurons, uses a linear activation function. The network was trained using the Adam optimizer (learning rate = 0.001) and the mean squared error (MSE) loss function over 1,000 epochs. To mitigate overfitting, early stopping was implemented based on validation loss, and training data were augmented with ±5% Gaussian noise in spectral intensity. The final training and validation loss curves exhibited stable convergence, confirming the robustness and generalization capability of the model. Computations were executed on a platform equipped with a GPU (NVIDIA GeForce RTX 3090) running CUDA 11.1. The evaluation metrics included the mean absolute error and accuracy rate.

## Data availability

The data supporting the findings of this study are contained primarily within the article and its Supplementary Information. Additional data are accessible from the corresponding author upon reasonable request.

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

## Acknowledgements

National Natural Science Foundation of China (Grant. 62305394); Distinguished Scholar of National Science Fund of China (No. 12225511) and Major Project (T2241002); National Natural Science Foundation of China (No. 12304362, 62375232); University Grants Committee /Research Grants Council of the Hong Kong Special Administrative Region, China [Project No. AoE/P-502/20, CRF Project: C5031-22G; C5078-24G, GRF Project: CityU11305223; CityU11300224; CityU11304925; CityU11305125]; City University of Hong Kong [Project No. 9380131].

## Author contributions

R.W., C.C., and D. P. T. conceived and designed the research. R.W. and D.Z. fabricated the metasensor chips. R.W. and D.Z. carried out the experiments. R.W., L.C., N.Z., D.L., R.J., X.Z., X.Y., L.Z., S.W., and X.L. analyzed and interpreted the data. R.W., C.C., and D.P.T. co-wrote the manuscript. All authors contributed to the results and commented on the manuscript.

## Competing interests

The authors declare no competing interests.
