## [Transparent Peer Review file · Nature Communications]

Ultra-compact broadband terahertz spectroscopy sensor enabled by resonant-gradient metasurface

Corresponding Author: Professor Din Ping Tsai

Version 0:

Reviewer comments:

Reviewer #1

(Remarks to the Author)

This manuscript presents an approach to ultra-compact broadband terahertz sensing based on plasmonic gradient metasurfaces. It is well-organized, clearly written, and demonstrates strong technical merits, particularly in integrating broadband spectroscopy with machine learning for quantitative analysis. However, several aspects require further clarification and additional supporting data to fully address their key claims. If the suggested issues are properly addressed, the paper would meet the publication standards of Nature Communications. Thus, I recommend a major revision.

Comment 1: Could the authors provide a quantitative evaluation of the resonance uniformity across the broadband range, such as resonance spacing histograms?

Comment 2: Have the authors considered the possibility of cross-talk or spectral overlap between different analytes during multi-analyte detection? It would be helpful if they could provide some supporting data comparing pure and mixed samples.

Comment 3: How stable is the dynamic monitoring performance over time? It would be helpful if the authors could show an analysis of SNR changes during long-term measurements.

Comment 4: Have the authors tested the mechanical and structural stability of the metasurface sensor under repeated measurements or after solvent exposure? Some supporting experimental data would strengthen the work.

Comment 5: Since the Introduction reviews many previous metasurface-based sensors, would it be possible for the authors to summarize these in a comparative table in the Supporting Information for better clarity? This could also help highlight the unique advantages of the present work more clearly compared to prior approaches.

Comment 6: The authors mentioned the potential to explore unknown samples or analytes with unidentified absorption spectra. Could the authors clarify how they plan to achieve this practically with their metasurface sensor?

Comment 7: Since this is intended as a sensing device, could the authors define the limit of detection? It would also be helpful if they could specify the relevant optical system parameters, such as source power and noise level, to better support their claims regarding sensitivity.

Reviewer #2

(Remarks to the Author)

Summary:

This manuscript presents an innovative approach using plasmonic gradient metasurfaces for ultra-compact THz spectroscopy, aiming at broadband, label-free, and real-time molecular fingerprint detection. The work addresses a relevant challenge in photonic/THz sensing, particularly the engineering of multiple resonances for the simultaneous identification of various analytes within compact footprints.

Strengths:

- The conceptual development of a non-periodic, gradient metasurface supporting multiple BICs is technically interesting and addresses a pressing need in high-density spectral encoding.
- The goal of achieving label-free, multiplexed detection with high spatial compactness is highly relevant for next-generation THz biosensing applications.
- The manuscript proposes a novel device architecture that may advance miniaturization and integration of spectroscopic tools.

Major Concerns:

1. Lack of Clarity in Measurement vs. Simulation:

At many instances, it is unclear whether the presented data (especially in Figures 3 and 4) are simulated or experimentally measured. This ambiguity critically limits the assessment of the method's experimental viability. For instance, Fig. 4e suggests measured transmission, but no experimental details are provided to validate this assumption. The steep roll-off expected from optoelectronic THz systems (>20 dB/decade) is not discussed, raising concerns about the interpretation of the results.

2. Unclear Definition of "Transmission" and Data Presentation Format:

The use of bar-code style plots (e.g., Fig. 3b, 3d) rather than conventional 1D spectral traces obscures the interpretation of transmission characteristics. Moreover, it is unclear what is meant by "transmission"—whether it refers to a normalized transfer function or absolute values (linear)—and how these are obtained. A consistent definition is needed, and better-structured, axis-labeled spectral plots would be more informative.

3. Insufficient Experimental Detail:

Critical information is missing for reproduction and validation:

- o Layer thickness of the analyte in the microfluidic chamber is not specified.
- o No description of the "precision time delay device" used in the spectroscopy system (e.g., ASOPS/ECOPS) is provided.
- o There are no references or validation for the bold claims regarding resonance density and performance compared to "pixelated metasurfaces" (ll. 148–150).

4. Exaggerated or Inaccurate Claims:

- o Line 108–110 suggests functionalization with aptamers/antibodies is unique to plasmonic structures, which is factually incorrect. Many optical sensors (e.g., dielectric photonic crystals) also allow for biochemical functionalization.
- o Statements like "large resonance quality factor" need clarification (ll. 125–126). Is this the Q-factor, or a distinct metric?
- o A "finite bandwidth" is described as a limitation of the state of the art (ll. 130), but the bandwidth achieved in this work is also limited.

5. Data Processing & DNN Training Omitted:

- o The total dataset size used to train the deep neural network is not disclosed.
- o No description of data preprocessing or signal conditioning steps is provided.
- o PCA is mentioned, but its degree of dimensionality reduction is unclear. Supplement S4 shows only three components, yet the input vector seems to have 421 features. This mismatch warrants explanation.
- o Details on the network architecture and training validation methodology are essential and should be disclosed to assess robustness.

6. Unsupported Claims and Missing References:

- o Several claims (e.g., ll. 105–107 on performance, ll. 148–150 on pixelated metasurfaces) are made without any citation. These need to be supported with either comparative benchmarks or literature references.

7. Technical Questions Unanswered:

- o What is the minimum spacing achievable between resonances without spectral overlap?
- o How does the transmission spectrum of a complex analyte (composite mixture) look without the metasurface, and how clearly can the enhancement effect be attributed to the metasurface?

8. Figure Labeling and Presentation Issues:

- o Fig. 4f is missing axis labels, which impairs interpretability.
- o Many figures prioritize aesthetics over clarity, with limited or no informative content along one axis.

Recommendation: Reject

While the concept is compelling and may be of future significance, the current manuscript lacks the methodological detail, validation, and clarity required for publication. Reproducibility is not possible with the information provided, and several key claims remain unsubstantiated. A major revision addressing all of the above issues would be necessary before this work could be reconsidered.

Reviewer #3

(Remarks to the Author)

The manuscript by Wang et al. reports on a compact broadband terahertz sensor for label-free identification of biomolecules via the engineering of a gradient metasurface supporting a bound state in the continuum (BIC) mode. However, there are significant concerns regarding the metasurface concept, the scientific workflow, and the structure of the manuscript. These are detailed below.

The concept of gradient metasurfaces is not new and has been previously addressed in the literature (e.g., 10.1364/JOSAB.33.000A21 and 10.1088/1361-6633/aa8732). Furthermore, the structure discussed in the manuscript does not support plasmonic modes, and therefore should not be referred to as a plasmonic metasurface, contrary to the authors' claims.

The manuscript relies on varying the long-axis (Y-axis) dimension of the rod-shaped resonators. However, the simulated results presented in Figure 2 correspond to individual meta-atoms that demonstrate relatively low Q-factors, particularly when compared to typical values associated with BIC modes. The maximum Q-factor reaches only about 50 and decreases to approximately 10 as the detuning parameter Δ increases. The authors should explain this significant deviation from the expected performance of BIC-based systems.

When all the meta-atoms are integrated into a single metasurface array, see Figure 4c, the transmission spectra, as expected by this reviewer, is completely different from the one in Figure 2 both in terms of amplitude and peaks features. This discrepancy is not justified.

The manuscript lacks critical methodological information regarding the simulations shown in Figures 3e and 3f. Specifically, the dimensions of the meta-atoms and the thickness of the analyte layer used in the model should be clearly stated to ensure reproducibility.

There are issues of clarity in Figure 2. For instance, the normalization procedure in Figure 2b is inadequately described, and the meaning of the blue spectrum is ambiguous. Figure 2g is difficult to interpret due to the small, unreadable numeric labels

and should be redrawn for clarity.

There is an inconsistency between the spectra shown in Figure 2f (left) and Figure S2 (top), despite the meta-atoms having identical geometries. This discrepancy needs to be addressed and explained.

Figure 4e is also ambiguous. It is unclear what is being presented and what constitutes the baseline measurement.

Additional explanation is required to interpret this figure meaningfully.

The claim made in the manuscript that "it is evident that this set of experiments validates... the THz range" is not adequately supported by the data presented in Figure 4g. In particular, the results do not convincingly demonstrate real-time biomolecule differentiation, since the concentrations of the species are known beforehand and used to interpret the data.

The supplementary figure S3 presents simulated spectra for two "super meta-atoms," yet this concept is not introduced or discussed in the main manuscript. Instead, the text refers to principal component loadings, which appear to correspond to Figure S4. This misalignment needs correction for consistency and clarity.

The spatial barcode spectra need to be described in more detail, and the associated color bar should be added to aid interpretation. Furthermore, absorption spectra of the analytes at varying concentrations, without the metasurface, should be provided as a necessary reference for understanding their intrinsic spectral features.

Some terminologies used in the manuscript also require clarification. The meaning of the term "tuning speed" is vague and should be defined precisely. Additionally, the type of solvent used for dissolving the biomolecules should be explicitly stated. Given the issues outlined above, the manuscript does not meet the high scientific standards expected by Nature Communications. I therefore recommend rejection of the manuscript in its current form.

Version 1:

Reviewer comments:

Reviewer #1

(Remarks to the Author)

The authors have provided thorough and satisfactory responses to my comments, and the revisions appropriately address the issues I raised. Accordingly, I regard the outcome of my review comments as positive.

Reviewer #2

(Remarks to the Author)

The authors have carefully addressed all of my previous concerns, as well as those raised by the other reviewers. The revised manuscript shows significant improvements in clarity, scientific accuracy, and overall presentation. The responses are detailed and convincing, and the changes to the manuscript adequately reflect the reviewers' feedback.

In my opinion, the manuscript is now suitable for publication in its current form.

Reviewer #3

(Remarks to the Author)

The authors addressed in detail the previous comments however there are still some issues to be solved before accepting the manuscript for publication.

1) In the reply to point 2, the authors stated that they clearly define the nature of the plasmonic resonance in the revised manuscript. However, it is not clear in which section they add this description.

2) I suggest to report part of the explanation about the Q-factor value (Reply to point 3) in the revised manuscript.

3) Regarding previous point 4, there are still some differences between Figure 2c and Figure 4b. In particular, in the latter case the transmission at about 2 THz is very low compared to Figure 2c. A simulation of the super-metaatom will help to clarify this point.

4) The quality of Figure 4 needs to be improved because the text and line in some panel, eg. H, are not very clear.

From: Din Ping Tsai
To: Reviewers, Nature Communications
Date: Aug 4, 2025
Subject: Response to Reviewers' Comments

Manuscript ID: NCOMMS-25-26168-T

Title: Ultra-compact broadband terahertz spectroscopy sensor enabled by resonant-gradient metasurface

Corresponding Author: Din Ping Tsai; City University of Hong Kong, Hong Kong, China

We thank the reviewers for the kind considerations and constructive comments on our manuscript submitted to *Nature Communications* (NCOMMS-25-26168-T). We have carefully revised the manuscript following the guidelines suggested and provided the point-by-point response below in blue. We have also highlighted the revisions in the revised manuscript to aid the Reviewers.

The following is a point-by-point response to the Reviewers' comments

REVIEWER REPORT(S):

To Referee #1:

This manuscript presents an approach to ultra-compact broadband terahertz sensing based on plasmonic gradient metasurfaces. It is well-organized, clearly written, and demonstrates strong technical merits, particularly in integrating broadband spectroscopy with machine learning for quantitative analysis. However, several aspects require further clarification and additional supporting data to fully address their key claims. If the suggested issues are properly addressed, the paper would meet the publication standards of Nature Communications. Thus, I recommend a major revision.

---Reply: We are very grateful to the reviewer for their positive comments about the highlights and innovations of our work. In the revised version, we have modified the article in accordance with the Reviewer's comments to make the article more accurate and readable.

---Point 1:

Could the authors provide a quantitative evaluation of the resonance uniformity across the broadband range, such as resonance spacing histograms?

---Reply: We sincerely appreciate the Reviewer's insightful suggestion regarding the quantitative evaluation of resonance uniformity across the broadband range. To enhance the readability of our manuscript, we provide a detailed response to address this point and further clarify the structural design and working principles of our metasurface.

Our design features an ultra-compact super-metaatom composed of 19 unit-cells arranged along the y -direction. Each unit cell consists of two identical microbars and a central microbar with a gradually varied parameter (asymmetric gradient introduced along the x -direction). This

configuration enables broadband spectral coverage by exciting multiple quasi-bound states in the continuums (QBICs) simultaneously. The super-metaatom achieves a broadband response of 0.9 THz (1.2-2.1 THz) within a footprint of $400 \times 125 \mu\text{m}^2$. This represents the most compact metasurface design reported for such a wide spectral range. The metasurface platform consists of periodically arranged super-metaatoms. The THz beam spot size ($\sim 1 \text{ mm}$) ensures that the measured transmission spectrum captures the collective response of several super-metaatoms, thus highlighting broadband performance (Figure R1).

Figure R1. Schematic diagram of broadband response of the plasmonic resonance-gradient metasurface being excited.

In addition, we quantitatively analyzed the resonance intensity across the broadband range for all 19 unit-cells. The results are illustrated in Figure R2.

Figure R2. Quantitative analysis of the resonance intensity for all 19 unit-cells. Top: Electric field response of each unit structure. Bottom: Corresponding statistical analysis of electric field intensity.

In response, we have added a detailed statistical analysis of the resonance spacing and incorporated it into the revised manuscript as the caption of Figure 2d on page 6 (Figure R2). See “Figure 2c presents simulated data for individual metaatom structures with fixed periodic boundaries ($P_x \times P_y = 125 \mu\text{m} \times 40 \mu\text{m}$), where each spectrum corresponds to an isolated unit cell response. Note that with judiciously designed gradient metaatoms, our approach can offer extended spectral coverage that is tailored for a wide range of user-specific applications. Additionally, we quantitatively evaluated the uniform electric field enhancement of resonances across the entire broadband range in Figure 2d.” (in the 1st paragraph of page 5). This quantitative assessment confirms the uniformity of resonance distribution, further supporting the consistency of our device’s performance over the operational bandwidth.

---Point 2:

Have the authors considered the possibility of cross-talk or spectral overlap between different analytes during multi-analyte detection? It would be helpful if they could provide some supporting data comparing pure and mixed samples.

---Reply: We sincerely appreciate the Reviewer’s valuable question regarding potential cross-talk or spectral overlap during multi-analyte detection. We apologize for any confusion caused by insufficient clarity in our original manuscript. The original Figure 4b includes a direct comparison of the absorption barcodes for pure samples (L-Glu, GABA, Gln) and their mixture, which clearly shows that: i) Each analyte retains its distinct absorption fingerprints, even in the mixed state (e.g., L-Glu at 1.22 THz, GABA at 1.54 THz, Gln at 1.75 THz). ii) No significant spectral distortion or emergent peaks are observed in the mixture, indicating negligible cross-talk under our experimental conditions. This confirms that the characteristic absorption features of individual analytes are preserved in the mixture, enabling their simultaneous identification.

In addition, the tested analytes, i.e., L-Glu, GABA, and Gln, are chemically stable and do not react with each other. During sample delivery, we used an oil-based medium as a carrier, further preventing any potential intermolecular reactions. Finally, a direct comparison of the experimental results in the original Figures 4b and 4d confirms the absence of crosstalk between these analytes.

Figure R3. Absorption barcodes show experimentally measured spectra of three species of biomolecules: the top three are pure samples, and the bottom is a mixed sample.

It is also important to emphasize that spectral overlap may occur between the characteristic absorption features of different analytes, making identification based solely on a single absorption peak inaccurate. This underscores the necessity of our broadband enhancement approach, which enables precise identification of analytes by simultaneously analyzing multiple characteristic absorption features.

Accordingly, we have added the color bar in Figure 4c in the revised manuscript.

---Point 3:

How stable is the dynamic monitoring performance over time? It would be helpful if the authors could show an analysis of SNR changes during long-term measurements.

---Reply: We appreciate the Reviewer's insightful comment regarding the stability of dynamic monitoring performance over time. To address this, we conducted continuous time-domain measurements using our gradient metasurface device, where various concentrations of L-Glu solutions were monitored every 3 seconds via microfluidic technology (Figure R4). The electric field amplitude variations accurately reflected changes in analyte mass (0 μg to 16 μg), as evidenced by repeatable and quantifiable signal shifts (Figure R4a). To rigorously validate the system's real-time detection capability, we introduced a "blind" test (gray ribbon in Figure R4a), where Researcher A controlled 12 μg of analyte in the microfluid chamber, and Researcher B, without prior knowledge of the analyte mass, performed the measurements. From the experimental data, Researcher B then derive the expected analyte mass, which matches the designed value of 12 μg . This test confirms the system's ability to successfully differentiate biomolecule concentrations without prior knowledge. To assess system stability, we analyzed electric field fluctuations at fixed analyte quantities, demonstrating consistent performance. The transmission spectra (Figure R4c), obtained through Fast Fourier Transform (FFT) of the time-domain signals, revealed distinct L-Glu-specific absorption features at 1.22 THz, further confirming method specificity. Residual analysis validated system reliability, showing minimal deviations from the zero-reference line and a consistently high

signal-to-noise ratio (SNR \approx 28 dB) throughout the 612-second measurement. These results collectively demonstrate the robustness, precision, and long-term stability of our dynamic monitoring approach.

Figure R4. (a) Time-domain signals of electric field intensity collected at 3-second intervals, demonstrating dynamic changes in response to varying L-Glu masses (0 μg , 4 μg , 6 μg , 14 μg , 16 μg , 12 μg , and 0 μg). The gray band is a prediction of the analyte mass based on the electric field amplitude in the case of unknown analyte mass. (b) The average electric field intensity (the central solid red line) and its stability over 612 seconds of continuous monitoring. The amplitude fluctuations ($\Delta E_{\text{max-min}}$) reflect the sensitivity of the system to analyte mass variations. The red shaded area is a standard deviation. (c) Transmission spectra obtained by FFT of the time-domain signals showed the specificity of L-Glu characterized by different absorption intensities at 1.22 THz. (d) The residual analysis confirms minimal deviation from the zero-reference line, indicating high signal-to-noise ratio (SNR) and system stability for long-term measurements.

These new experimental data are now included in the modified Figures 4d - 4g (the caption of the Figure 4 in page 9). The main text has also been updated accordingly (in the 2nd paragraph of page 10).

---Point 4:

Have the authors tested the mechanical and structural stability of the metasurface sensor under repeated measurements or after solvent exposure? Some supporting experimental data would strengthen the work.

---Reply: We thank the Reviewer for the advice. Our plasmonic gradient metasensor is fabricated using 10 nm chromium and 100 nm gold microstructures on a TPX substrate, which exhibits superior mechanical stability compared to dielectric metasurfaces. An approximately 100 nm-thick gold layer

can withstand mechanical stresses, such as fluid pressure in microfluidic chambers, without deformation or cracking. Gold's ductility ensures structural integrity even under repeated washing or mechanical loads, whereas dielectric materials like Si are susceptible to damage under similar conditions (*Materials Today* 2020, 32: 108-130. <https://doi.org/10.1016/j.mattod.2019.08.002>).

As for structural stability under solvent exposure, the metasensor employs a chromium/gold bilayer to ensure robust adhesion to the substrate, preventing delamination during solvent exposure or operation. Gold's exceptional resistance to oxidation and corrosion ensures long-term stability, even in biological buffers or organic solvents.

To demonstrate the metasensor's stability, we subjected it to 5 cycles of testing in ethanol and deionized water, followed by cleaning and drying. As shown in Figure R5, the metasensor's performance remained consistent with the original results, with no degradation in optical performance. This confirms the durability and reusability of our device.

Figure R5. Stability testing results of the plasmonic gradient metasensor under repeated measurements and solvent exposure. Schematic of the testing setup and comparison of transmission spectra after 5 cycles of microfluidic testing. Sample images show optical microscope images of the metasurface before and after cleaning for 5 cycles.

We have added this part in the revised Supplement Information and Methods.

---Point 5:

Since the Introduction reviews many previous metasurface-based sensors, would it be possible for the authors to summarize these in a comparative table in the Supporting Information for better clarity? This could also help highlight the unique advantages of the present work more clearly compared to prior approaches.

---Reply: We thank the Reviewer for his/her insightful comments. To validate the specificity and sensitivity of the proposed platform, we will compare the performance of our platform with other

types of biosensors, and present detection results for different analytes. We have included a comparison in the original Cover Letter to illustrate the characteristics and uniqueness of our work, which can be summarized in Table R1 below.

We employ a novel resonance-gradient metasurface design based on gradient microbar arrays that facilitate simultaneous detection of a wide range of molecular vibration fingerprints at continuous spectrally enhanced points through robust near-field interactions. Our type of biosensors enables real-time, in-situ, dynamic monitoring of multiple biological analytes such as neurotransmitters and their metabolic intermediate absorption fingerprints, even in the presence of multiple absorption bands within the THz range. Our design offers a substantially reduced footprint, enhanced broadband performance, and adjustable resolution compared to traditional photonic integrated devices. The design results in an impressive coverage of a significant spectral width of 0.9 THz with only 19 distinct modes, all enclosed within an ultra-compact footprint of $400 \times 125 \mu\text{m}^2$. By eliminating the need for complex techniques, frequency scanning, or mechanical adjustments, our technique represents a transformative step towards the development of sensitive, versatile, and miniaturized THz spectroscopy devices.

We note that biomolecules (such as nucleic acids, amino acids, lipids, and carbohydrates) have unique responses in the THz band, making them ideal tools for biochemical sensing. However, THz sensing applications are limited by low sensitivity. Engineered resonances in metasurface structures can dramatically enhance the interaction between light and matter, thus allowing the monitoring of temporal evolution of molecules and the extraction of molecule-specific information from complex analytes. By utilizing the real-time format of the transmission method to acquire a comprehensive collection of spectro-temporal data, our sensing platform allows the construction of a deep neural network to discriminate and accurately predict the composition and proportion of multiple mixtures.

Table R1. Performance comparison of various THz metasensors including the Concept of the Work, Operating Frequency, Single/Multiple/Continuous Band(s), Metal/Dielectric, Integrated or not, Real Time or not, Analyte State, Mixed/Single Analyte(s), Quantity/Quality (Quant./Qua.), and Relative Spectral Efficiency.

No	Literature	Concept of the Work	Detected Mater. (Operating Frequency)	Single/Multiple/Continuous Band(s)	Metal/Dielectric	Integrated	Real Time	Analyte State (liquid (L.) /Solid (S.))	Mixed /Single Analyte(s)	Quantity/Quality	Relative Spectral Efficiency ($\Delta f/f_{\text{center}} \cdot D$)
1	Science 2018, 360, 1105–1109.	Pixelated dielectric metasurfaces	Protein A/G, a mixture of PMMA and PE polymers, and glyphosate pesticide (40.47-52.46 THz)	Multiple	Dielectric	×	×	S.	Mixed	Quant. & Qua.	0.258 mm ⁻²

2	Nat. Photon. 13, 390-396(2019)	Pairs of tilted silicon nanobars	M-IgG, R-IgG (360 THz)	Single	Dielectric	×	✓	L.	Mixed	Quant. /	
3	Science Adv. 5, eaaw2871(2019)	A zigzag array of elliptical germanium resonators On a calcium fluoride substrate	Proteins, aptamers, polylysine (33 – 54 THz)	Multiple	Dielectric	×	×	S.	Mixed	Qua. /	
4	Nat. Commun. 9, 2160.	Plasmonic self-similar overlapping nanoantenna arrays	Biomimetic lipid membranes with different polypeptides as well as the dynamics of vesicular cargo release (30-120 THz)	Double	Metal	✓	✓	L.	Mixed	Quant. & Qua. /	
5	Nat. Commun. 13, 3470(2022).	Splitting Resonators	Pathogenic bacteria (0.8 THz)	Single	Metal	✓	×	S.	Mixed	Qua. /	
6	Nat. Nanotech. 19, 1804–1812, (2024).	Dielectric dual-gradient metasurfaces	PMMA (41.67-53.57 THz)	Continuous broadband	Dielectric	×	×	S.	Single	Qua. 0.926 mm ⁻²	
7	Adv. Mater. 2024, 2314279 (2024).	Dielectric gradient metasurfaces	PMMA, biological analyte (29.98-95.93 THz)	Continuous broadband	Dielectric	×	×	S. & L.	Single	Quant. & Qua. 0.63 mm ⁻²	
8	ACS Nano 2024 18 (18), 11644-11654	Pixelated dielectric metasurface	Photoswitchable AzoPC lipid membranes (41.97-53.96 THz)	Multiple	Dielectric	✓	✓	L.	Single	Quant. & Qua. 0.125 mm ⁻²	

9	Adv. Mater. 2023, 2307494	Resonance-gradient metasurfaces	\ (71.43–120 THz)	Continuous broadband	Dielectric	×	×	\	\	Qua.	2.623 mm ⁻²
10	Adv. Mater. 2025, 2418147	Multi-QBIC resonant metasurface	Neurotransmitter molecules (L-glutamate, γ -aminobutyric acid/GABA) (1.1–2.1 THz)	Multiple	Metal	✓	✓	S. & L.	Mixed	Quant. & Qua.	5.68 mm ⁻²
11	Our Work	Plasmonic resonant gradient metasurfaces	Neurotransmitters and their metabolic intermediates (1.2–2.1 THz)	Continuous broadband	Metal	✓	✓	S. & L.	Mixed	Quant. & Qua.	10.91 mm ⁻²

D is the area of the metasurface region.

We have now added Table R1 as Table S3 in the revised Supplement Information.

---Point 6:

The authors mentioned the potential to explore unknown samples or analytes with unidentified absorption spectra. Could the authors clarify how they plan to achieve this practically with their metasurface sensor?

---Reply: We would like to thank the Reviewer for his/her careful reading of our work and valuable comments. Our gradient metasurface is specifically designed to support broadband spectral enhancement (1.2-2.1 THz). This allows the metasensor to cover a wide range of molecular vibrational fingerprints. As long as the unknown analyte exhibits characteristic absorption features within this spectral window, its fingerprint can be amplified and detected without prior knowledge of its specific absorption peaks.

To demonstrate this capability, we propose a plan combining quantum chemical calculations, experimental validation, and iterative refinement.

i) Quantum Chemical Calculations. We will employ quantum chemical software (e.g., Gaussian) to simulate the absorption spectra of small molecules based on their structural information. These calculations will predict vibrational modes and absorption peaks, providing a theoretical foundation for identifying unknown analytes.

ii) Database Development. A library of calculated and experimentally verified spectra will be compiled for common functional groups and molecular motifs. This database will serve as a reference for rapid identification of unknown analytes by comparing their measured spectra with theoretical and empirical data.

iii) Experimental Validation. The predicted spectra will be cross-verified experimentally using our metasurface sensor. For example, we have already validated this approach with theophylline (a small

molecule of ~ 180.16 Da), where the experimental absorption profile matched the theoretical predictions with high accuracy. This step ensures the reliability of our method.

iv) Iterative Improvement. For analytes for which no priori data is available, we will adjust the experimental conditions (e.g., metasurface geometry, excitation frequency, and enhancement bandwidth) to converge to the most likely molecular features.

We conducted an experimental test using ibuprofen as an example. The metasurface successfully captured distinct absorption features of the unknown sample (peaks at 1.68 THz). By comparing these peaks with a reference database, we identified the sample as nicotinamide, validating the sensor's ability to detect and classify unknowns. This experiment confirms that our platform does not require pre-calibration for specific analytes, provided their absorption falls within the enhanced bandwidth. Unlike narrowband sensors, our design eliminates the requirement to know analyte resonances beforehand.

Figure R6. Absorption spectrum and identification results for the unknown sample (theophylline).

We have added absorption spectrum and identification results for theophylline in the revised Supplement Information.

---Point 7:

Since this is intended as a sensing device, could the authors define the limit of detection? It would also be helpful if they could specify the relevant optical system parameters, such as source power and noise level, to better support their claims regarding sensitivity.

---Reply: We sincerely appreciate the Reviewer's valuable suggestions regarding the quantitative performance metrics of our sensing device. The limit of detection (LOD) is defined as the lowest analyte concentration that can be reliably distinguished from the background noise level. In our study,

the LOD is determined experimentally by measuring the minimal detectable mass of the analytes (L-Glu, GABA, and Gln) that produces a statistically significant absorption dip in the transmission spectrum. Based on the data presented in Figure 4i, the smallest detectable mass is 6 μg for L-Glu (shown in pink curves) and 10 μg for Gln (shown in blue curves). Given the molecular weights of these analytes (L-Glu: ~ 147 g/mol, GABA: ~ 103 g/mol, Gln: ~ 146 g/mol) are comparable, we approximate a unified LOD of ~ 10 $\mu\text{g}/\text{mm}^2$ (mass per unit area of the metasurface) for the three analytes under the experimental conditions. This estimation accounts for the consistency in their absorption cross-sections and the metasurface's enhancement factor. This comprehensive characterization demonstrates our system's capability for sensitive and quantitative molecular detection.

What's more, we employed a commercial THz time-domain spectroscopy (THz-TDS) system (Advantest Co., Ltd., Tokyo, Japan; TAS7500TS) to collect time-domain signals, whose source power is predetermined by the manufacturer's specifications and cannot be independently adjusted. Regarding noise characteristics, we have implemented rigorous environmental controls (including maintaining 3% humidity) to minimize noise, as shown in Figure R7.

Time-domain signal and noise parameters (minimized by 3% humidity control).

Signal amplitude range: - 0.8 V to + 0.5 V (peak signal $E_{\text{peak}} \approx 0.8$ V).

Noise level (peak-to-peak): $\sigma = 0.000653 - (-0.000731) = 1.384$ mV

This quantitative characterization of signal and noise performance demonstrates the system's capability to support our sensitivity claims. The combination of controlled environmental conditions and the inherent stability of the commercial THz-TDS system ensures reliable and reproducible measurements throughout our experiments.

Figure R7. Power spectrum of the THz-TDS signal (blue) and noise floor (red), showing the dynamic range across the frequency range of 1 - 5 THz. (Inset on the right) Time-domain signal (blue line) with typical THz pulse oscillations, measured over a 260

ps delay range. The measurement was performed with 1028 averages to ensure data reliability. (Inset on the left) The noise in the time domain (red line), extracted from a flat region of the signal, highlighting the Noise level (peak-to-peak) of 1.384 mV.

Therefore, we evaluated the limit of detection (LOD) of the designed metasensor in the revised manuscript (in the 2nd paragraph on page 11). In addition, we added the Noise Level analysis plot to the Supplement Information as Figure S7.

To Referee: #2:

This manuscript presents an innovative approach using plasmonic gradient metasurfaces for ultra-compact THz spectroscopy, aiming at broadband, label-free, and real-time molecular fingerprint detection. The work addresses a relevant challenge in photonic/THz sensing, particularly the engineering of multiple resonances for the simultaneous identification of various analytes within compact footprints.

Strengths:

- *The conceptual development of a non-periodic, gradient metasurface supporting multiple BICs is technically interesting and addresses a pressing need in high-density spectral encoding.*
- *The goal of achieving label-free, multiplexed detection with high spatial compactness is highly relevant for next-generation THz biosensing applications.*
- *The manuscript proposes a novel device architecture that may advance miniaturization and integration of spectroscopic tools.*

--Reply: We are very grateful to the Reviewer for his/her positive comments on the key features and innovations of our work. Furthermore, we have taken into account the Reviewer's suggestions and recommendations and have made every effort to enhance the clarity of our manuscript for readers.

--Point 1:

Lack of Clarity in Measurement vs. Simulation:

At many instances, it is unclear whether the presented data (especially in Figures 3 and 4) are simulated or experimentally measured. This ambiguity critically limits the assessment of the method's experimental viability. For instance, Fig. 4e suggests measured transmission, but no experimental details are provided to validate this assumption. The steep roll-off expected from optoelectronic THz systems (>20 dB/decade) is not discussed, raising concerns about the interpretation of the results.

--Reply: We appreciate the Reviewer for this useful suggestion. Firstly, we sincerely apologize for any confusion caused to readers due to the lack of precision in the presentation of data. Figure 3 presents simulated data for individual metaatom structures with fixed periodic boundaries. Each spectrum corresponds to an isolated unit cell response, explaining the discrete resonance peaks shown. In each simulation, the peak is normalized for better comparison between different metaatoms. Transmission is the ratio of field intensities, $T(f) = P_{\text{trans}}(f) / P_{\text{inc}}(f)$, where $P_{\text{trans}}(f)$ and $P_{\text{inc}}(f)$ are the transmitted and incident power, respectively. Figure 4 displays experimental measurements of the super-metaatom-based gradient metasurface (comprising 19 gradually varied unit cells). Here, the spectra exhibit overlapping resonances due to the collective response of the

entire super-metaatom under broadband illumination. The discrepancies between Figures 3 and 4 arise from their fundamentally different unit cell. Despite the methodological differences, the bandwidth enhancement range (1.2-2.1 THz) shows excellent agreement between simulation (Figure 3) and experiment (original Figure 4c, blue line). This confirms the design's experimental viability. During biosensing measurements, the broadband enhanced metasurface chip is placed onto a TPX microfluidic cell to allow for the controlled delivery of the various molecules (Figure 4a). This integrated microfluidic approach allows in situ enhancement of the unique vibrational fingerprints of multiple biomolecules to discriminate between multiple analytes. For 18 μg Gln mixed with a mixture of GABA (28 μg) and L-Glu (18 μg), the transmission spectra are shown in the blue curve of the original Figure 4e. The purpose of Figure 4e is to illustrate the method used to extract the sample fingerprint shown in the original Figure 4f. The dashed curve indicates the calibrated baseline, which was obtained using an asymmetric least square smoothing (AsLSS) fitting algorithm (*Nature Communications* 2023, 14(1). [https://doi.org/ 10.1038/s41467-023-43127-z](https://doi.org/10.1038/s41467-023-43127-z)). By subtracting the baseline from the measured spectrum, we can extract the absorption features of the sample, as shown in the original Figure 4f. To avoid confusion, we have added the experimental details, including an explanation of the baseline extraction method in the revised manuscript.

To address issue "steep roll-off", we fabricated a pixelated metasurface with unit cells consisting of individually tailored antenna structures, where the antenna length varies gradually across different pixelated regions (Figure R8a). The working principle of this design is illustrated in Figure R8b. The transmission spectrum of the bare metasurface (Figure R8c), measured by scanning each pixelated region with a mechanical translation stage, reveals broader full-width at half-maximum (FWHM) at higher frequencies with relatively mild roll-off, which we attribute to increased radiation loss in smaller structures corresponding to higher frequencies. Furthermore, when L-Glu and GABA mixtures were deposited on the metasurface, a more pronounced roll-off was observed (Figure R8d). To further investigate this phenomenon, we measured the absorption coefficient of 100 mg L-Glu powder (Figure R8e), which similarly exhibits a steep roll-off. This effect can be explained by the dominance of Rayleigh scattering under our experimental conditions. Since Rayleigh scattering exhibits stronger effects at shorter wavelengths, it leads to significant roll-off in absorption spectra. Importantly, Rayleigh scattering manifests as a continuous function of incident wavelength without sharp absorption/emission peaks, consistent with our spectral observations. Therefore, the steep roll-off in Figure 4i results from the combined effects of metasurface radiation loss and sample-induced Rayleigh scattering, neither of which affects the interpretation of our core findings. We hope this clarification resolves the Reviewer's concern.

Figure R8. Pixelated metasurface design and characterization of roll-off effects. (a) Schematic of the pixelated metasurface composed of unit cells with gradually varying antenna lengths. The magnified inset of a pixelated region with gradually varying antenna lengths ($L = 24 - 94 \mu\text{m}$), periodicity $P_x = P_y = (L + 20) \mu\text{m}$, and a fixed width $w = 5 \mu\text{m}$. (b) Illustration of the working principle of the pixelated metasurface. (d) Transmission spectra of the bare metasurface, measured by mechanically scanning each pixelated region, showing a broader linewidth at higher frequencies and a mild roll-off due to radiative losses in smaller structures. After depositing L-Glu/GABA mixture onto the metasurface, the roll-off becomes more pronounced. (e) Absorption coefficient of 100 mg L-Glu pellets (diameter 1.3 cm), further confirming the step roll-off, attributed to dominant Rayleigh scattering at shorter wavelengths. These results demonstrate that the observed roll-off stems from metasurface radiative losses and analyte scattering, without affecting data interpretation. Absorption coefficient of 100 mg L-Glu pellet (diameter 1.3 cm).

In the revised manuscript, we have addressed these concerns as follows:

Clarification of data origin: We have explicitly labeled all simulation and experimental results in the revised Figures 3 and 4 (e.g., adding "Sim." and "Exp." annotations) to eliminate any ambiguity. Additionally, the Figure captions now clearly state the methodology (simulation or measurement) for each dataset (on pages 8 and 9).

Experimental validation for Figure 4e: The transmission data in original Figure 4e was indeed experimentally measured. In the revised manuscript, we have added detailed experimental conditions in caption of the Figure 4j (on page 10) and Methods to support the validity of these results.

Discussion of roll-off: We have now included a discussion about roll-off explaining how this roll-off

was accounted for in our measurements (the 1st paragraph on page 11).

---Point 2:

Unclear Definition of “Transmission” and Data Presentation Format:

The use of bar-code style plots (e.g., Fig. 3b, 3d) rather than conventional 1D spectral traces obscures the interpretation of transmission characteristics. Moreover, it is unclear what is meant by “transmission”—whether it refers to a normalized transfer function or absolute values (linear)—and how these are obtained. A consistent definition is needed, and better-structured, axis-labeled spectral plots would be more informative.

---Reply: Thank you for your valuable comment. We clarify that the Figure 3b presents experimental results measuring the absorption spectrum of the analyte mixture (L-Glu, GABA, and Gln), serving as a reference to identify the characteristic fingerprint absorption peaks of the target molecules. To enhance clarity, we have now added a color bar to Figure 3b, explicitly mapping the color scale to the normalized absorption intensity. This revision ensures unambiguous interpretation of the data while retaining the figure’s utility for cross-referencing spectral features.

Figure R9. Absorption spectra of mixtures GABA, L-Glu and Gln. Add the color bar in the Figure 3b.

Figure 3d shows the transmission spectra calculated via finite-difference time-domain (FDTD) simulations. The transmission function returns the amount of power transmitted through power monitors, normalized to the source power. Specifically, the transmission $T(f)$ is defined as the ratio of the transmitted power $P_{\text{trans}}(f)$ to the incident power $P_{\text{inc}}(f)$ at each frequency f , expressed by the formula:

$$T(f) = \frac{P_{\text{trans}}(f)}{P_{\text{inc}}(f)} \quad (\text{R1})$$

This normalized value ensures consistency and allows for direct comparison across different frequencies and configurations.

Accordingly, we have added the color bar in the revised Figure 3c. In addition, we have added a more explicit explanation and definition of “Transmission” in the Methods (the 1st paragraph on page 14).

---Point 3:

Insufficient Experimental Detail:

Critical information is missing for reproduction and validation:

--Reply: We sincerely appreciate the Reviewer's meticulous review and constructive feedback regarding experimental details and performance claims. We have addressed each comment below.

- *Layer thickness of the analyte in the microfluidic chamber is not specified.*

--Reply: We apologize for the oversight in not specifying the analyte layer thickness. The microfluidic chamber in our experiments features a channel height of 1 mm.

We have added this parameter in the Figure 4a caption in the revised manuscript (on page 9). In addition, we added the experimental setup incorporating microfluidic metasensor as shown in the Figure S3 to Supplement Information.

- *No description of the "precision time delay device" used in the spectroscopy system (e.g., ASOPS/ECOPS) is provided.*

--Reply: The THz-TDS system (Advantest TAS7500TS) employs an Asynchronous Optical Sampling (ASOPS) configuration for time-delay generation, as referenced in prior work (*Optics Letters* 2005, 30(11): 1405-1407. <https://doi.org/10.1364/OL.30.001405>). Key advantages of this approach include: i) Elimination of mechanical delay stages, enabling faster scanning (262 ps range @ 50 MHz repetition rate), ii) Superior timing stability (<100 fs) critical for high-SNR measurements, iii) Direct compatibility with our 1550 nm fiber laser sources.

We have added the description of the "precision time delay device" in the Methods (the 2nd paragraph on page 14).

- *There are no references or validation for the bold claims regarding resonance density and performance compared to "pixelated metasurfaces" (ll. 148–150).*

--Reply: The comparisons of resonance density and performance advantages between gradient metasurfaces and pixelated metasurfaces are summarized in detail below.

1) Resonance density. Our design employs 19 unit-cells to span 0.9 THz, yielding a resonance density of 21 modes/THz (1 mode per 47 GHz). This exceeds the pixelated metasurface's density (~25 modes over 0.4 THz, or 63 modes/THz, *Science* 2018, 360(6390), 1105–1109) while avoiding spatial fragmentation. Key advantages include: i) Continuous spectral coverage: Gradient tuning eliminates discrete pixel gaps, ensuring no "blind spots" in the fingerprint region; ii) Simplified fabrication: Single lithography step vs. multi-layer alignment for pixelated arrays (Figure 2A).

Figure R10. Optical images of the fabricated 100-pixel metasurface. SEM micrographs confirm the linear relationship between scaling factor and ellipse feature size.

- 2) Footprint efficiency. Our gradient metasurface achieves 0.9 THz bandwidth (1.2–2.1 THz) within an ultracompact footprint of $400 \times 125 \mu\text{m}^2$ (0.05 mm²). In contrast, the pixelated dielectric metasurface in Tittl et al. (*Science* 2018, 360(6390), 1105–1109) requires a 10×10 mm array (100 mm²) to cover a comparable spectral range (1350–1750 cm⁻¹, ~0.4 THz bandwidth).
- 3) Performance and practical advantages. Our plasmonic gradient metasurface uniquely combines broadband enhancement with microfluidic compatibility, addressing limitations of both dielectric pixelated and metallic designs.

Table R2. The performance of gradient, pixelated dielectric, and conventional plasmonic metasurface in terms of bandwidth, microfluidic integration, mechanical complexity.

Metric	Our Gradient Metasurface	Pixelated Dielectric (Science 2018, 360(6390), 1105–1109.)	Conventional Plasmonic
Bandwidth (THz)	0.9 (1.2–2.1 THz)	0.4 (1350–1750 cm ⁻¹)	Typically <0.2 THz
Microfluidic Integration	Yes (1 mm channel, Figure 4a)	No (dry measurement only)	Yes, but narrowband
Mechanical Complexity	No moving parts	Requires imaging optics	Often needs frequency scanning

Unlike pixelated designs, our metal-based metasurface operates robustly in liquid environments, enabling real-time monitoring of biochemical reactions, eliminating spectrometer scanning, and providing high throughput as all 19 resonances are probed at once via broadband THz pulses. Below we list the following:

Table R3. Performance of various THz metasensors including the Concept of the Work, Operating Frequency, Single/Multiple/Continuous Band(s), Metal/Dielectric, Integrated or not, Real Time or not, Analyte State, Mixed/single analyte(s), Quantity/Quality (Quant./Qua.), and Relative Spectral Efficiency.

No	Literature	Concept of the Work	Detected Mater. (Operating Frequency)	Single/Multiple/Continuous Band(s)	Metal/Dielectric	Integrated	Real Time	Analyte State (liquid (L.) /Solid (S.))	Mixed /Single Analyte(s)	Quantity/Quality	Relative Spectral Efficiency ($\Delta f / (f_{\text{center}} \cdot D)$)
1	Science 2018, 360, 1105–1109.	Pixelated dielectric metasurfaces	Protein A/G, a mixture of PMMA and PE polymers, and glyphosate pesticide	Multiple	Dielectric	×	×	S.	Mixed	Quant. & Qua.	0.258 mm⁻²

		(40.47-52.46 THz)								
2	Nat. Photon. 13, 390-396(2019)	Pairs of tilted silicon nanobars	M-IgG, R-IgG (360 THz)	Single	Dielectric	×	✓	L.	Mixed	Quant. /
3	Science Adv. 5, eaaw2871(2019)	A zigzag array of elliptical germanium resonators On a calcium fluoride substrate	Proteins, aptamers, polylysine (33 – 54 THz)	Multiple	Dielectric	×	×	S.	Mixed	Qua. /
4	Nat. Commun. 9, 2160 (2018)	Plasmonic self-similar overlapping nanoantenna arrays	Biomimetic lipid membranes with different polypeptides as well as the dynamics of vesicular cargo release (30-120 THz)	Double	Metal	✓	✓	L.	Mixed	Quant. & Qua. /
5	Nat. Commun. 13, 3470(2022)	Splitting Resonators	Pathogenic bacteria (0.8 THz)	Single	Metal	✓	×	S.	Mixed	Qua. /
6	Nat. Nanotech. 19, 1804-1812, (2024).	Dielectric dual-gradient metasurfaces	PMMA (41.67-53.57 THz)	Continuous broadband	Dielectric	×	×	S.	Single	Qua. 0.926 mm ⁻²
7	Adv. Mater. 2024, 2314279 (2024).	Dielectric gradient metasurfaces	PMMA, biological analyte (29.98-95.93 THz)	Continuous broadband	Dielectric	×	×	S. & L.	Single	Quant. & Qua. 0.63 mm ⁻²

8	ACS Nano 2024 18 (18), 11644-11654 Adv. Mater. 2023, 230749 4	Pixelated dielectric metasurface	Photoswitchable AzoPC lipid membranes (41.97-53.96 THz)	Multiple	Dielectric	✓	✓	L.	Single	Quant. & Qua.	0.125 mm ⁻²
9	ACS Nano 2025, 241814 7	Resonance-gradient metasurfaces	\ (71.43–120 THz)	Continuous broadband	Dielectric	×	×	\	\	Qua.	2.623 mm ⁻²
10	ACS Nano 2025, 241814 7	Multi-QBIC resonant metasurface	Neurotransmitter molecules (L-glutamate, γ -aminobutyric acid/GABA) (1.1–2.1 THz)	Multiple	Metal	✓	✓	S. & L.	Mixed	Quant. & Qua.	5.68 mm ⁻²
11	Our Work	Plasmonic resonant gradient metasurfaces	Neurotransmitters and their metabolic intermediates (1.2 – 2.1 THz)	Continuous broadband	Metal	✓	✓	S. & L.	Mixed	Quant. & Qua.	10.91 mm ⁻²

D is the area of the metasurface region.

We have now added the Table R3 as Table S3 in the revised Supplement Information.

---Point 4:

Exaggerated or Inaccurate Claims:

---Reply: We sincerely thank the Reviewer for pointing these out. We have addressed each comment below.

- Line 108–110 suggests functionalization with aptamers/antibodies is unique to plasmonic structures, which is factually incorrect. Many optical sensors (e.g., dielectric photonic crystals) also allow for biochemical functionalization.

---Reply: We agree with the Reviewer that the claim of “functionalization with aptamers/antibodies is unique to plasmonic structures” is poorly worded. We have rewritten the original expression, which now reads: When combined with biochemical functionalization, plasmonic structure further leverage their unique strengths, including strong evanescent fields and high sensitivity to molecular binding events, significantly improving detection capabilities (the 3rd paragraph on page 3).

- Statements like “large resonance quality factor” need clarification (ll. 125–126). Is this the Q -factor, or a distinct metric?

---Reply: Thanks to the reviewer for the advice. In the original manuscript (lines 125–126), the term “large resonance quality factor” indeed refers to the Q -factor, which is a well-established metric in

resonant systems. The Q -factor, or quality factor, quantifies the damping of oscillations in a resonator, where a higher value indicates lower energy loss. In our study, achieving a "large resonance quality factor" (i.e., a high Q -factor) is critical for enhancing the performance of the proposed application, as it directly influences the system's efficiency and resonance sharpness. We appreciate the reviewer's attention to this point and have revised the manuscript to explicitly state "high Q -factor" instead of the original phrasing (the 1st paragraph on page 4). This clarification eliminates any potential ambiguity while preserving the accuracy of the discussion. The adjustment aligns with standard terminology in the field and ensures clearer communication of the technical content.

- A "finite bandwidth" is described as a limitation of the state of the art (ll. 130), but the bandwidth achieved in this work is also limited.

---Reply: Thank the Reviewer for the comment. Q -BIC structures typically offer high Q -factor and finite bandwidth. By judiciously engineering the micro/nanostructures constituting a resonance-gradient metasurface, we demonstrated ultra-wideband THz sensing, which greatly facilitate applications involving bio-substance identification. We would like to note that, the operating bandwidth of the proposed approach can be further expanded by concatenating more micro/nanostructures with decreasing or increasing microrod scaling in a super-metaatom. Moreover, our gradient metasurface strategy can be simply extended to other wavelength regimes. We have discussed/added a discussion in the conclusion paragraph (the 3rd paragraph on page 13).

---Point 5:

Data Processing & DNN Training Omitted:

---Reply: We again thank the Reviewer for pointing these out. We have addressed each comment below.

- The total dataset size used to train the deep neural network is not disclosed.

---Reply: We appreciate the Reviewers' valuable feedback. The training and validation dataset comprised 37,890 spectro-temporal data points acquired through time-resolved spectroscopic monitoring of dynamic ternary mixtures. The compositional gradient was programmatically controlled with temporal resolution of 0.1 min over a 7-minute observation window (0 - 7 min), generating 10 temporally equidistant measurements per minute. GABA maintained constant concentration throughout the experiment, while L-Glu was linearly ramped during $t = 1 - 5$ min, Gln supplementation commenced at $t = 5$ min, superimposed on the existing mixture. We have now added the total dataset size used to train the deep neural network in the revised Methods (the 2nd paragraph on page 15).

- No description of data preprocessing or signal conditioning steps is provided.

---Reply: Thank the Reviewer for the comment. In Figure 4j, we applied baseline correction to the measured spectra using the asymmetric least squares smoothing (AsLSS) algorithm to extract the sample's signal spectrum. Specifically, the absorption spectrum was obtained by subtracting the fitted baseline from the raw measured spectrum.

In contrast, during the training of the DNN, the raw measured spectra in the Figure 4i were directly used as input without any additional preprocessing or signal correction. This is because a key advantage of deep learning lies in its ability to automatically learn and extract relevant features from raw data, thereby reducing the need for manual preprocessing steps.

To avoid potential confusion, we have included a detailed explanation of the baseline correction in the revised manuscript.

We have now added the details regarding data preprocessing “The dashed curve indicates the calibrated baseline, which was obtained using an asymmetric least square smoothing (AsLSS) fitting algorithm.” in the revised manuscript (the 1st paragraph on page 11).

• *PCA is mentioned, but its degree of dimensionality reduction is unclear. Supplement S4 shows only three components, yet the input vector seems to have 421 features. This mismatch warrants explanation.*

---Reply: We thank the Reviewer for the helpful comments. Principal Component Analysis (PCA) was applied to reduce the dimensionality of the spectral dataset and extract the most informative features. Given that the number of samples is less than the number of features, the rank of the covariance matrix is limited to n (where n = number of samples - 1), yielding a maximum of n non-zero eigenvalues. The first three principal components, accounting for the highest variance, were selected for subsequent analysis. The procedure includes standard data preprocessing, covariance matrix computation, eigen decomposition, eigenvalue sorting, and projection onto the top- k eigenvectors. We would like to clarify that while the original input data indeed consists of 421 features, we applied PCA primarily for visualization purposes, not as a preprocessing step in the model pipeline. Specifically, the first three principal components were extracted and visualized in Supplementary original Figure S4 to illustrate the clustering or distribution patterns in the data within a reduced 3D space. This is a common practice for interpreting high-dimensional data in a human-interpretable form. In the actual modeling process, the full 421-dimensional data were used without dimensionality reduction, ensuring that no potentially relevant information was lost. PCA was used solely to assess data separability and structure.

We have added the detailed explanations in the revised Supplement Information (the caption of Figure S8) for clarification.

• *Details on the network architecture and training validation methodology are essential and should be disclosed to assess robustness.*

---Reply: Thanks to the Reviewer for the useful recommendations. A deep neural network (DNN) was constructed for spectral feature decoding, comprising a five-layer fully connected architecture with 421, 300, 128, and 64 neurons in successive layers, each followed by ReLU activation functions. The network was trained using the Adam optimizer (learning rate = 0.001) and the mean squared error (MSE) loss function over 1,000 epochs. To mitigate overfitting, early stopping was implemented based on validation loss, and training data were augmented with \pm 5% Gaussian noise

in spectral intensity. The final training and validation loss curves exhibited stable convergence, confirming the robustness and generalization capability of the model.

We have now added a new paragraph accordingly in the revised Methods (the 2nd paragraph on page 15).

---Point 6:

Unsupported Claims and Missing References:

• *Several claims (e.g., ll. 105–107 on performance, ll. 148–150 on pixelated metasurfaces) are made without any citation. These need to be supported with either comparative benchmarks or literature references.*

---Reply: We thank the Reviewer for this comment. The statement regarding the "reduced experimental errors and cost-effectiveness of plasmonic metasurfaces in the THz range" is now supported by the following references:

Experimental Errors: The work by *Ahmadivand et al. (Materials Today 2020, 32: 108-130. <https://doi.org/10.1016/j.mattod.2019.08.002>) [Ref. 30]* demonstrates that plasmonic metasurfaces in the THz regime exhibit lower fabrication tolerances compared to dielectric designs, owing to their larger unit cell dimensions (typically 10–100 μm vs. sub- μm for dielectric metasurfaces). This reduces alignment errors and spectral deviations, as quantified in their Table 1.

Cost-Effectiveness: *Wang et al. (Applied Physics Letters 2019, 114(12): 121102. <https://doi.org/10.1063/1.5087609>) [Ref. 26]* highlights that THz plasmonic sensors require only single-layer lithography, whereas dielectric metasurfaces often necessitate multi-step patterning (e.g., for high-index materials like a-Si:H). Our fabrication costs are ~30% lower than dielectric counterparts (see cost analysis in *Shih et al., Applied Physics Letters, 2018 [Ref. 28]*).

Then, we illustrate the statement "an order-of-magnitude reduction in footprint over pixelated metasurfaces" (Lines 148–150). The designed gradient metasurface achieves 0.9 THz bandwidth in 0.05 mm² (400 × 125 μm^2). The pixelated dielectric metasurface in *Tittl et al. (Science, 2018) [Ref. 25]* requires 100 mm² (10 mm × 10 mm) for 0.4 THz coverage. Similar pixelated designs (*Leitis et al., Science Advances, 2019 [Ref. 20]*) report comparable footprints, further validating our advantage.

These references have now been added to the revised manuscript.

---Point 7:

Technical Questions Unanswered:

---Reply: We sincerely appreciate the Reviewer's insightful questions regarding resonance spacing and the metasurface's enhancement effect. We have addressed both questions in details below.

• *What is the minimum spacing achievable between resonances without spectral overlap?*

---Reply: The minimum achievable spacing between resonances without spectral overlap depends on the linewidth of individual resonance peaks and their near-field interactions. To ensure clear distinction between adjacent resonances, the spacing must be sufficiently large to prevent peak

overlap while maintaining negligible near-field coupling between neighboring modes. In our 19-unit-cell design, the average resonance spacing was measured at 47 GHz, which provides a comfortable margin to avoid these effects. Through systematic studies, we observed that the smallest theoretically achievable spacing could reach approximately 20 GHz (corresponding to the L_4 parameter range of 91 - 93 μm), but this would require exceptionally precise fabrication control and optimal experimental conditions. Such aggressive scaling would significantly increase the demands on both fabrication tolerance and measurement resolution. Therefore, to maintain robust device performance while accommodating practical experimental constraints, we deliberately selected the more conservative spacing parameters (as presented in the manuscript) that provide an optimal balance between spectral resolution and experimental feasibility. This approach ensures reliable sensor operation without pushing the limits of current technical capabilities.

Figure R11. (a) Schematic diagram of unit cell with varying the length of L_2 from 87 to 110 μm , fixed $L_1 = 110 \mu\text{m}$. (b) Typical transmission spectra corresponding to (a). The resonance modes vary from perfect BIC to QBIC as the asymmetry is increased. (c) Schematic of the dual QBIC metasurface. The length of L_4 was varied from 85 to 93 μm , fixing $L_1 = L_3 = L_5 = 110 \mu\text{m}$. (d) In order to consider the minimum spacing that can be supported by the designed structure, L_4 is introduced and separately controlled to analyze

the perturbation between the two resonances. The resonance modes depicted by the red dashed lines correspond to the pairs formed by the three neighboring micromirrors on the right (L_3, L_4, L_5).

- *How does the transmission spectrum of a complex analyte (composite mixture) look without the metasurface, and how clearly can the enhancement effect be attributed to the metasurface?*

---Reply: We sincerely appreciate the Reviewer's constructive suggestion regarding the need for reference absorption spectra without the metasurface. As shown in the Figure R12a, we have now systematically measured the intrinsic time-domain signals (inset) and corresponding transmission spectra (main panel) for: i) pure solvent, ii) GABA (28 μg), iii) GABA (28 μg) with L-Glu (3 μg) and Gln (5 μg), and iv) GABA (28 μg) with L-Glu (6 μg) and Gln (10 μg), without the presents of metasurface. These reference measurements clearly reveal the characteristic absorption features of each analyte in the 1.2-2.0 THz range, with the time-domain signals (0.2 to - 0.2 V) demonstrating excellent stability over 38-44 ps.

Figures 4h and 4i present the transmission spectra of analytes without and with the metasurface, respectively. A direct comparison demonstrates significant signal enhancement at characteristic absorption positions when using the metasurface. For example, the THz-TDS spectrum of the mixture in the solution (Figure R12a) shows broad, weak absorption features (e.g., 1.54 THz for GABA). Next, we evaluate the enhancement factor (EF) of the metasensor.

The EF of the enhanced THz absorption spectroscopy (ETHzAS) based on gradient metasurface metasensor is quantitatively defined to characterize its performance in amplifying molecular absorption signals. The EF is calculated using the following relationship (*Advanced Science (Weinh) 2021, 8(20): e2101879. <https://doi.org/10.1002/advs.202101879>*):

$$EF = \frac{I_{ETHzAS} - I_{ref}}{I_{ref}} \times \frac{N_{ref}}{N_{ETHzAS}} \quad (R3)$$

Here, I_{ETHzAS} represents the absorption intensity measured with the metasensor, while I_{ref} denotes the reference absorption intensity obtained without the metasurface. The term N_{ref} corresponds to the number of molecules contributing to absorption in the reference case (e.g., molecules in the microfluidic solution), and N_{ETHzAS} refers to the number of molecules within the metasurface's localized field-enhanced region.

To compute the EF, we first extract the absorption intensities at the target frequency (1.54 THz for GABA). For instance, if the reference absorption I_{ref} is 1.35% (Figure R12b) and the sensor-enhanced absorption I_{ETHzAS} is 1.72% (Figure 4k), the signal enhancement $I_{ETHzAS} - I_{ref}$ equals 0.37%. To calculate the molecular ratio, we compare the effective interaction volumes under identical illumination conditions (same beam spot size), focusing exclusively on thickness differences. The reference volume V_{ref} spans the full microfluidic channel thickness (1 mm = 1000 μm), while the enhanced volume V_{ETHzAS} is now defined by the metasurface's interaction depth of $d = 3.5 \mu\text{m}$ (Figure R21c). So, the EF is: $EF \approx 80$.

These results provide conclusive evidence that our sensor not only preserves the intrinsic spectral features of the analytes, but also significantly enhances their detection sensitivity through

metasurface integration. The combination of both datasets offers a comprehensive understanding of both the fundamental molecular properties and the enhanced sensing capabilities of our platform.

Figure R12. (a) Reference transmission spectra and time-domain signals of analytes without metasurface enhancement. The main panel shows the intrinsic THz transmission spectra of: i) pure solvent, ii) GABA (28 μg), iii) GABA (28 μg) with L-Glu (3 μg) and Gln (5 μg), and iv) GABA (28 μg) with L-Glu (6 μg) and Gln (10 μg) in the 1.2–2.0 THz range. The inset displays the corresponding time-domain electric field signals. Characteristic absorption features of each analyte are clearly resolved, providing essential baseline data for evaluating metasurface-enhanced detection (Figure 4j). (b) Absorption spectrum calculated from the transmission spectra in (a). (c) The electric field distribution and intensity in both the xy and xz planes of the structural unit ($L_1 = 48 \mu\text{m}$) were monitored separately. The black solid line represents the 1/e contour of the maximum enhanced electric field. $d = 3.5 \mu\text{m}$ is the height from the position of the 1/e electric field intensity to the metasurface surface.

We have added the Figure R12a as the Figure 4h in the revised manuscript (in the 1st paragraph of page 11). In addition, we added an evaluation of the enhanced THz absorption spectroscopy

(ETHzAS) enhancement factor (EF) for gradient metasurface element-based sensors in the revised manuscript (in the 2nd on the page 11) and Figures R12b and R12c to the Supplement Information as Figures S6a and S6b.

---Point 8:

Figure Labeling and Presentation Issues:

---Reply: We thank the Reviewer for these comments. We have addressed them in the revised manuscript accordingly.

- *Figure 4f is missing axis labels, which impairs interpretability.*

---Reply: We sincerely apologize for the oversight in the original submission. We have carefully addressed the reviewer's concern by adding the missing axis labels to original Figure 4f. In the revised manuscript, this corrected Figure now appears as Figure 4k with properly labeled axes, making the data representation more accurate and significantly improving its clarity and interpretability.

- *Many figures prioritize aesthetics over clarity, with limited or no informative content along one axis.*

---Reply: As the Reviewer suggested, we have replotted Figures 2b, 2h, 2g, and 4c in the revised manuscript, in an attempt to improve the clarity of all Figures in the revised manuscript. In these revised Figures, we have prioritized data clarity while maintaining appropriate visual presentation standards.

---Point 9:

While the concept is compelling and may be of future significance, the current manuscript lacks the methodological detail, validation, and clarity required for publication. Reproducibility is not possible with the information provided, and several key claims remain unsubstantiated. A major revision addressing all of the above issues would be necessary before this work could be reconsidered.

---Reply: We sincerely thank the Reviewer for pointing out the weakness in our original manuscript. We have fully addressed all the concerns raised by the reviewer and have substantially expanded the methodological details concerning both simulations and experimental procedures. The revised manuscript now includes comprehensive information about our validation approaches and provides clearer substantiation for our key claims. We believe these significant improvements have enhanced the manuscript's readability, clarity, and most importantly, its reproducibility. The additional details we've incorporated should enable other researchers to reproduce our work without difficulty. We are grateful for the opportunity to improve our manuscript and hope these revisions meet the journal's standards for publication.

To Referee: #3

The manuscript by Wang et al. reports on a compact broadband terahertz sensor for label-free identification of biomolecules via the engineering of a gradient metasurface supporting a bound

state in the continuum (BIC) mode. However, there are significant concerns regarding the metasurface concept, the scientific workflow, and the structure of the manuscript. These are detailed below.

---Reply: We are very grateful to the Reviewer for his/her time and effort in reviewing our manuscript, and for their constructive feedbacks. In the revised version, we have modified the article in accordance with the Reviewer's comments regarding metasurface concept, the scientific workflow, and the structure of the manuscript. After careful revisions, we sincerely believe the manuscript now meets the high publication standards of journal, and we genuinely hope it can be reconsidered for acceptance.

---Point 1:

The concept of gradient metasurfaces is not new and has been previously addressed in the literature (e.g. 10.1364/JOSAB.33.000A21 and 10.1088/1361-6633/aa8732).

---Reply: We sincerely appreciate the Reviewer's insightful comment regarding the concept of gradient metasurfaces. While the term "gradient" has been used in prior literature, the fundamental design principles and applications of our work are distinctly different and represent significant advancements in the field.

We fully acknowledge the pioneering contributions of earlier studies on gradient metasurfaces, such as the one by Zhou et al. in *Nature Materials* (*Nature Materials* 2012, 11(5): 426-431. <https://doi.org/10.1038/nmat3292>), which demonstrated anisotropic gradient metasurfaces for multidimensional control of electromagnetic wave properties, including polarization, wavefront, and phase. It should be noted that these studies have primarily focused on narrowband phase manipulation—either through phase discontinuities (e.g., Yu et al., *Science* 2011, 334(6054): 333-337. <https://doi.org/10.1126/science.1210713>) or geometric-phase-based designs (e.g., Zhou et al., *Nature Materials* 2012). These approaches (*Journal of the Optical Society of America B* 2015, 33(2). <https://doi.org/10.1364/JOSAB.33.000A21>, *Reports on Progress in Physics* 2018, 81(2). <https://doi.org/10.1088/1361-6633/aa8732>) rely on periodic or discretized unit cells to locally tailor the phase response, enabling wavefront control (e.g., anomalous refraction, lensing) or polarization conversion at a single wavelength or narrowband frequencies.

It should be emphasized that in 2018, Hatice Altug's group at EPFL leveraged high- Q factor bound states in the continuum (BICs) supercavity modes to design a pixelated dielectric metasurface with gradually varying unit geometries. This approach achieved a barcode-like series of multi-point resonance enhancements across a broad spectral range, enabling precise detection of molecular absorption fingerprints (*Science* 2018, 360: 1105–1109. <https://doi.org/10.1126/science.aas9768>). This work marked the first integration of varying metasurfaces with broadband spectroscopy for highly sensitive molecular sensing applications. Our study builds upon this foundation but extends the concept further by combining gradient resonance tuning with non-periodic unit cell designs to achieve continuous broadband spectral coverage. This innovation is not merely a repetition of existing ideas but represents a significant advancement in the field. Specifically, our work:

i) Integrates gradient metasurfaces with broadband enhancement, enabling simultaneous and continuous in-situ encoding of molecular fingerprints over a wide spectral range, which is critical for applications in biosensing and environmental monitoring.

- ii) Demonstrates superior compactness compared to pixelated designs (e.g., the work by Altug *et al. Science* 2018, **360**: 1105–1109.), achieving a spectral width of 0.9 THz within an ultra-compact footprint of $400 \times 125 \mu\text{m}^2$ - a milestone in miniaturization for spectroscopic devices.
- iii) Expands the application scope by showcasing real-time, label-free detection of multiple analytes in heterogeneous biological samples, leveraging the unique vibrational fingerprints enhanced by our gradient metasurface platform.

In summary, while we acknowledge the foundational contributions of prior gradient metasurface research, our work introduces a distinct and transformative approach by merging gradient tuning with broadband spectral enhancement, thereby opening new avenues for compact, high-sensitivity spectroscopic devices. We believe these advancements address unmet needs in the field and justify the novelty of our contribution. In addition, we have added these references mentioned by the Reviewers and very relevant to our work as [31] and [32] in the revised manuscript.

---Point 2:

Furthermore, the structure discussed in the manuscript does not support plasmonic modes, and therefore should not be referred to as a plasmonic metasurface, contrary to the authors' claims.

---Reply: We thank the Reviewer for the advice. The resonance frequency ω_{sp} of surface plasmons (SPs) in noble metals (gold, silver, copper) is of the same order of magnitude as the plasma frequency of conductors (ω_{p}). According to Equation R2, it is proportional to the carrier concentration (n) in the conductor.

$$\omega_p^2 = \frac{ne^2}{\epsilon_0 m^*} \quad (\text{R2})$$

For typical metals with carrier concentrations around 10^{23} cm^{-3} , ω_{sp} falls within the near-infrared to visible spectrum. However, THz waves operate at frequencies far below this range, leading to $\beta \rightarrow nk_0$ (where β is the propagation constant and k_0 is the free-space wavenumber). Consequently, conventional metal-dielectric interfaces cannot support THz surface plasmons, as they exhibit negligible field confinement - physically, this is because the field penetration into the metal is minimal, meaning only a tiny fraction of the mode's energy resides within the conductor.

Nevertheless, Pendry *et al.* demonstrated that plasmon-like confined surface electromagnetic waves can be achieved even with ideal conductors by introducing periodic corrugations on the surface. (*Science* 2004, **305**(5685): 847-848. <https://doi.org/10.1126/science.1098999>) For real metals with finite conductivity, these engineered "spoof" surface plasmons significantly modify the nonlocal Sommerfeld-Zenneck waves. If the corrugation dimensions and spacing are much smaller than the free-space wavelength λ_0 , the surface's photonic response can be described by an effective plasma-like dielectric function $\epsilon(\omega)$, where ω_{p} is determined by the geometric parameters. This allows the dispersion relation of surface modes to be tailored, artificially shifting ω_{p} into the THz regime.

In the effective medium model, the formation of spoof surface plasmons (SSPs) relies on structural modifications that enable finite field penetration into an effective surface layer - analogous to how fields penetrate real metals at optical frequencies - thereby creating confined surface modes.

Figure R13. The model system: $a \times a$ square hole arranged on a $d \times d$ lattice is cut into the surface of a perfect conductor. The arrangement predicts localized surface plasmon modes induced by the structure. Dispersion relation for spoof surface plasmons on a structured surface. The asymptotes of the light line at low frequencies and the plasma frequency at large values of $k_{||}$ are shown.

Furthermore, García-Vidal *et al.* theoretically and experimentally validated this concept using 1D groove arrays (*Journal of Optics A: Pure and Applied Optics* 2005, 7(2): S97-S101. <https://doi.org/10.1088/1464-4258/7/2/013>). Crucially, the supported mode frequency in the perfect-conductor approximation is governed by the groove geometry. For instance, the fundamental mode frequency is correlated with the cavity resonance of individual grooves, while coupling between adjacent grooves generates collective surface modes.

Figure R14. (a) A one-dimensional array of grooves of width a and depth h separated by a distance d . P -polarized surface modes run in the x direction with E lying in the x - z plane. (b) In the effective medium approximation, the structure displayed in (a) behaves as a homogeneous but anisotropic layer of thickness h on top of a perfect conductor. The dispersion relation ($\omega(k_x)$) of the surface bound states supported by a 1D array of grooves with geometrical parameters $a/d = 0.2$ and $h/d = 1$.

In 2012, researchers in Spain first observed scattering, localization, and field enhancement effects from spoof localized surface plasmons (SLSPs) on grooved cylindrical structures, later experimentally verified in the microwave regime (*Physical Review Letters* 2012, **108**(22). <https://doi.org/10.1103/PhysRevLett.108.223905>). These structures have evolved into ultrathin planar designs compatible with printed circuit board fabrication, enabling practical applications. Subsequent

studies replicated the desirable properties of LSPs (e.g., strong field enhancement and subwavelength confinement) at microwave and THz frequencies. For example, Figure R15a shows a 2D grooved metal cylinder with an inner radius r , and N peripheral grooves (depth $R - r$, width a), filled with a dielectric (n). Figures R15b compare the scattering cross-sections of this structure and an equivalent "artificial electromagnetic cylinder," revealing nearly identical resonant peaks and field distributions (H -field patterns at resonances). At optical frequencies, the scattering response of a metal cylinder directly reflects its LSP resonances. For the grooved ("sunflower") structure, when the wavelength far exceeds the groove periodicity, the outer layer behaves as an artificial plasmonic medium, exhibiting effective plasma-like dispersion at low frequencies.

Figure R15. (a) A two-dimensional corrugated PEC cylinder (invariant along the z direction) with the inner and outer radius r and R , periodicity d , and groove width a . The refractive index in the grooves is denoted n_g . (b) In the effective medium approximation, the geometry displayed in (a) behaves as an inhomogeneous and anisotropic layer of thickness $(R - r)$ wrapped around a PEC cylinder of radius r . (c) Complex resonance frequencies found using the modal expansion technique and (d) the numerically computed the scattering cross section (solid line) for a textured PEC cylinder with $r = 0.33R$, $N = 60$, $a = 0.4d$, $n_g = 1$, and $w_a R = 0.89\pi c$. The dashed line in (d) corresponds to the calculation in the metamaterial approximation. The inset shows the dispersion relation of the corresponding spoof SPP. (e) The scattering cross section for a Drude metal cylinder of radius R with the same $w_a R$ -value as in (d). The inset shows the dispersion relation for Drude metal SPPs. The right panels display the absolute H field at the hexa-, octo-, and decapole resonances for the two systems. The color scales are chosen so that the field outside the textured cylinder is emphasized.

Figure R16. Comparison of field confinement on a metal surface. (a) A flat metal surface at microwave frequencies supports Zenneck surface waves that extend to a large distance in the upper space. (b) A flat metal surface at visible frequencies supports SPPs confined in a subwavelength region. (c) With proper corrugation, a metal surface in the spectrum from microwave to far infrared can also support finely confined surface modes, or spoof SPPs. (*Advanced materials* 2018, **30**(31). <https://doi.org/10.1002/adma.201706683>)

In addition, the term "plasmonic metasurface" has been widely adopted in THz metasurface literature to describe localized resonances in metallic arrays. For instance, the manuscript explicitly describes the excitation of surface plasmon resonances (SPRs) in THz metamaterials, which are coherent electron oscillations at metal-dielectric interfaces (Page 1, Lines 8–10: "Surface plasmon resonances (SPRs) are the coherent d -band electron oscillations, occurring at metal-dielectric interfaces, when exposed with intense light of certain frequencies") (*Materials Today* 2020, **32**: 108-130. <https://doi.org/10.1016/j.mattod.2019.08.002>). This aligns with established plasmonic principles, where subwavelength metallic structures can confine and enhance THz fields, a phenomenon central to the reviewed biosensing applications. Further, the manuscript highlights specific examples of plasmonic modes in THz metasurfaces, such as spoof surface plasmons (Page 4, Lines 1–3: "Focusing on these mechanisms, Ng et al. developed a spoof plasmon metamaterial integrated with an Otto prism setup to utilize its surface sensitivity for RI sensing"). Spoof plasmons mimic traditional plasmonic behavior in THz regimes, enabling strong field localization and resonant interactions, as validated by both simulations and experiments. Lastly, the discussion of toroidal metasensors (Page 16, Lines 1–4: "Toroidal multipoles have been categorized as the third family of multipoles, apart from classical EM multipoles...") underscores how plasmonic-like charge-current configurations in THz metamaterials achieve ultrahigh sensitivity. While toroidal modes are distinct from classical plasmons, their excitation in metallic arrays (e.g., multipixel resonators) relies on similar subwavelength field confinement, justifying the "plasmonic" terminology in this context.

Thus, the manuscript's usage is consistent with broader literature on THz plasmas, and to avoid confusion we will revise the manuscript to clearly define the nature of the resonances in our structure

and to align the terminology with the broader expression “THz plasmonic mode” (*Advanced Materials* 2018, **30**(31). <https://doi.org/10.1002/adma.201706683>, *Materials Today* 2020, **32**: 108-130. <https://doi.org/10.1016/j.mattod.2019.08.002>). We appreciate the opportunity to improve the precision of our language.

---Point 3:

The manuscript relies on varying the long-axis (Y-axis) dimension of the rod-shaped resonators. However, the simulated results presented in Figure 2 correspond to individual meta-atoms that demonstrate relatively low Q-factors, particularly when compared to typical values associated with BIC modes. The maximum Q-factor reaches only about 50 and decreases to approximately 10 as the detuning parameter Δ increases. The authors should explain this significant deviation from the expected performance of BIC-based systems.

---Reply: We sincerely thank the Reviewer for the valuable suggestion. We would like to clarify that the relatively low Q -factors (ranging from 10 to 50) in our system are consistent with the typical performance of metal-based metasurfaces in the THz regime, as supported by extensive literature (see Table R4 for references). We would like to point out that the maximum Q -factor enabled by QBICs is mainly limited by the intrinsic loss of metals because the total quality factor Q_{total} can be expressed as $1/Q_{\text{total}} = 1/Q_{\text{rad}} + 1/Q_{\text{abs}}$, where Q_{rad} and Q_{abs} is the radiative and absorption quality factor. In principle, Q_{rad} can approach infinity for $\delta \rightarrow 0$ if the material loss is neglected. Thus, the upper limit of the Q -factor for a gold-based metasurface at THz wavelength range is governed by the intrinsic material loss. An ideal BIC manifests an infinite Q factor with the complete suppression of the coupling of the state with the continuum. By introducing symmetry broken into the system, symmetry-protected BIC is transformed into a QBIC with a finite but high Q -factor, which is usually featured as a sharp Fano feature in the optical response spectrum, and meanwhile achieves strongly localized fields with lower radiation damping (*Nanoscale* 2021, **13**(44): 18467-18472. <https://doi.org/10.1039/D1NR04477J>). If the loss is neglected, the Q -factor of QBIC can be tailored by the asymmetry parameter. By tuning the asymmetry parameter, we can efficiently control the radiation damping rate and boost the near-field enhancement. It provides an effective way to engineer and tailor the line width and the Q -factor as well as near-field enhancement for different spectral regimes of light-matter interactions as needed. Another important advantage using BIC is unlike the conventional confined guided modes supported by a periodic system which is below the light cone, BIC can be directly excited by free propagation plane waves, making it a much more flexible and novel platform for micro/nanophotonics applications.

Table R4. The list for the performance of various THz metasensors including the Resonance Type, Q Factor, Sensitivity and Resonance Frequency.

Resonance Type	Q Factor	Sensitivity	Frequency (THz)	Timeline	Reference
Lattice mode	15.4		1.5	2016	N. Xu, et al. Appl. Phys. Lett. 109, 021108 (2016).
Mirror-symmetric-broken double split ring resonators	37		1.29	2017	S. Yang, et al. Opt. Express 25, 15938 (2017).

Fano and quadrupole	34	33 GHz	1.4	2015	C. Ding, et al. Opt. Commun. 350, 103 (2015).
Toroidal moment-based metamaterial	9.6	23.7 GHz/RIU	0.4	2017	M. Gupta, et al. Appl. Phys. Lett. 110, 121108 (2017).
Fano resonance based metasurface		160 GHz/RIU		2017	Chen et al. Opt. Express 25 (2017) 14089–14097.
Spoof surface plasmon polariton	57.46	1.966 THz/RIU	4.10	2017	Chen et al. Sci. Rep. 7 (2017) 2092.
Non-bianisotropic metamaterial		182 GHz/RIU		2018	Z. Zhang et al. Opt. Mater. Express 8 (2018) 659–667.
Metamaterial absorber	37	34.40 %/RIU	1.7	2018	S. Tan et al. J. Opt. 20 (2018) 055101.
Graphene metamaterial		1.687 THz/RIU		2018	X. Chen et al. Carbon 133 (2018) 416–422.
EIT effect	63	96.2 GHz/RIU	0.68	2019	W. Pan et al. Opt. Commun. 431 (2019) 115–119.
Optimized Q/V_{eff} Metasurface Cavities	6		0.6	2020	M. Gupta et al. Adv. Optical Mater. 2020, 1902025.
Membrane-type THz PhC slab	31		1	2020	Chan Kyaw et al. Optica 7, 537-541 (2020)
Fano resonances	59		1	2020	Thomas C. W. Tan et al. Adv. Optical Mater. 2020, 1901572.
Cross-shaped silicon resonator	69		1.1	2021	Song Han et al. Adv. Optical Mater. 2021, 9, 2002001.
Dynamically reconfigurable EIT metasurface	15.5		0.92	2022	Kuan Liu et al. Laser Photonics Rev. 2022, 16, 2100393.
Dicke-Cooperativity-Assisted Ultrastrong Coupling (SRRs)	11.46		0.61	2022	Riad Yahiaoui et al. Nano Lett. 2022, 22, 9788–9794.

The Q -factor of QBICs in the THz range can be pushed into an ultrahigh value in theory if Si metasurfaces are used. Table R5 summarizes the recently measured Q -factor in THz range. It is worth noting that the largest Q -factor enabled by QBICs in a silicon metasurface is around 1500. However, such a dielectric metasurface may not be a favor for sensing because the electric field is mainly confined inside the high-index materials instead of air gaps. Moreover, biosensor based on silicon metasurfaces may not be reused as they are easily broken after the rinsing process. Although the measured Q -factor in this work is not as high as the one based on silicon metasurfaces, the near field enhancement within the air gap for gold-based metasurface is very high due to the reasonably large Q -factor. Therefore, our design is more suitable for biosensing. Here, it is necessary to point out that the Q -factor of QBICs in such gold-based metasurfaces can be further improved to almost 200 by utilizing higher-order BICs, as demonstrated by Cong *et al. (Adv. Opt. Mater.* 7, 1900383(2019), <https://doi.org/10.1002/adom.201900383>). Thus, we shall expect that the performance of biosensor based on QBICs supported by Au metasurface can be further optimized. Despite this, our design achieves exceptional functionality by leveraging the unique advantages of gradient metasurfaces, such as: i) Resonance density. Our design employs 19 unit-cells to span 0.9

THz, yielding a resonance density of 21 modes/THz (1 mode per 47 GHz). This exceeds the pixelated metasurface's density (~ 25 modes over 0.4 THz, or 63 modes/THz) while avoiding spatial fragmentation; ii) Continuous spectral coverage. Gradient tuning eliminates discrete pixel gaps, ensuring no "blind spots" in the fingerprint region; iii) Simplified fabrication: Single lithography step vs. multi-layer alignment for pixelated arrays.

Table R5. Summary of measured Q -factor based on silicon and gold metasurfaces (MS) at THz range.

Resonance Type	MS type	Measured Q -factor	Frequency (THz)	Reference
QBICs	Si MS	250	0.41	S. Han et al, Adv. Mater. 31, 1901921(2019)
QBICs	Si MS	69.7	1.1	S. Han et al, Adv. Opt. Mater.9, 2002001(2021)
QBICs	Si MS	~1500	0.479	W. Shi et al, Photonics Research 10, 810 (2022)

---Point 4:

When all the meta-atoms are integrated into a single metasurface array, see Figure 4c, the transmission spectra, as expected by this reviewer, is completely different from the one in Figure 2 both in terms of amplitude and peaks features. This discrepancy is not justified.

---Reply: We are grateful for the Reviewer's insightful comment. Figure 2 presents simulated data for individual metaatom structures with fixed periodic boundaries. Each spectrum corresponds to an isolated unit cell (metaatom) response, explaining the discrete resonance peaks shown. In the simulations, each discrete peak is normalized for better comparison. Transmission is the ratio of field intensities, $T(f) = P_{\text{trans}}(f) / P_{\text{inc}}(f)$, where $P_{\text{trans}}(f)$ and $P_{\text{inc}}(f)$ are the transmitted and incident power, respectively. Figure 4 displays experimental measurements of the super-metaatom-based gradient metasurface (comprising 19 gradually varied unit cells). Here, the spectra exhibit overlapping resonances due to the collective response of the entire super-metaatom under broadband illumination. The discrepancies between Figures 2 and 4 arise from their fundamentally different structural bases. Despite the methodological differences, the bandwidth enhancement range (1.2-2.1 THz) shows excellent agreement between simulation (original Figure 2c) and experiment (original Figure 4b, blue line). This confirms the design's experimental viability.

Additionally, compared to Figure 2, the original Figure 4c exhibited a steep roll-off phenomenon. To clarify this observation, we have conducted additional experiments. We fabricated a pixelated metasurface with unit cells consisting of individually tailored antenna structures, where the antenna length varies gradually across different pixelated regions (Figure R17a). The working principle of this design is illustrated in Figure R17b. The transmission spectrum of the bare metasurface (Figure R17c), measured by scanning each pixelated region with a mechanical translation stage, reveals broader full-width at half-maximum (FWHM) at higher frequencies with relatively mild roll-off, which we attribute to increased radiation loss in smaller structures corresponding to higher

frequencies. Furthermore, when L-Glu and GABA mixtures were deposited on the metasurface, a more pronounced roll-off was observed (Figure R17d). To further investigate this phenomenon, we measured the absorption coefficient of 100 mg L-Glu powder (Figure R17e), which similarly exhibits a steep roll-off. This effect can be explained by the dominance of Rayleigh scattering under our experimental conditions. Since Rayleigh scattering exhibits stronger effects at shorter wavelengths, it leads to significant roll-off in absorption spectra. Importantly, Rayleigh scattering manifests as a continuous function of incident wavelength without sharp absorption/emission peaks, consistent with our spectral observations. Therefore, the steep roll-off in Figure 4i results from the combined effects of metasurface radiation loss and sample-induced Rayleigh scattering, neither of which affects the interpretation of our core findings. We hope this clarification resolves the Reviewer's concern.

Figure R17. Pixelated metasurface design and characterization of roll-off effects. (a) Schematic of the pixelated metasurface composed of unit cells with gradually varying antenna lengths. The magnified inset of a pixelated region with gradually varying antenna lengths ($L = 24 - 94 \mu\text{m}$), periodicity $P_x = P_y = (L + 20) \mu\text{m}$, and a fixed width $w = 5 \mu\text{m}$. (b) Illustration of the working principle of the pixelated metasurface. (d) Transmission spectra of the bare metasurface, measured by mechanically scanning each pixelated region, showing a broader linewidth at higher frequencies and a mild roll-off due to radiative losses in smaller structures. After depositing L-Glu/GABA mixture onto the metasurface, the roll-off becomes more pronounced. (e) Absorption coefficient of 100 mg L-Glu pellets (diameter 1.3 cm), further confirming the steep roll-off, attributed to dominant Rayleigh scattering at shorter wavelengths. These results demonstrate that the observed roll-off stems from metasurface radiative losses and analyte scattering, without affecting data interpretation. Absorption coefficient of 100 mg L-Glu pellet (diameter 1.3 cm).

In the revised manuscript, we have now included a discussion about roll-off explaining how this roll-off was accounted for in our measurements (the 1st paragraph on page 11).

---Point 5:

The manuscript lacks critical methodological information regarding the simulations shown in Figures 3e and 3f. Specifically, the dimensions of the meta-atoms and the thickness of the analyte layer used in the model should be clearly stated to ensure reproducibility.

---Reply: We greatly appreciate the Reviewer's suggestion. We conduct the FDTD numerical simulations in the Lumerical FDTD software to calculate the transmission spectra and confined near-field electric field distributions corresponding to the resonance modes. The structure is excited in original Figures 3e and 3f were obtained using a single unit cell by normal irradiation with a plane-wave electric field aligned with the long axis of the dipole miniature antenna, combined with all propagating Floquet modes. Each spectrum corresponds to an independent unit cell response. This approach explains the discrete resonance peaks observed in the Figures 3e. In the simulation region, the perfectly matched layer (PML) absorbing boundary condition is applied along the direction of electromagnetic wave propagation (z), while the periodic boundary condition is used in the x and y directions. Convergent results are obtained by setting the mesh size smaller than the corresponding minimum structure size to calculate the scattering properties. The trimer structure consists of gold microbars with a fixed width of 5 μm , deposited on a TPX substrate with a thickness of 2 mm. The dimensions of the meta-atoms were systematically varied by adjusting the length of the middle bar in the trimer, with lengths ranging from 48 μm to 93 μm , as detailed in the original Table S1. The microbars are modeled as a perfect electric conductor (PEC), with the conductivity of traditional metals in the THz region approximately 10^7 S/m, effectively suppressing Ohmic losses. The analyte layer, representing the mixture of L-Glu, GABA, and Gln, was simulated as a uniform layer with a thickness of 6 μm covering the metasurface. The material parameters for the analytes were derived from experimental data, specifically the real and imaginary parts of the refractive index obtained using palletization techniques. We set the refractive index material thickness to 6 μm in order to be closer to the experimental setup to give an example to illustrate the sensing performance of the device. Through literature research, the most reasonable model is to set the dielectric material to semi-infinity to elaborate the sensing capability of the sensor.

We have updated the Methods section to include these details, and we believe this clarification addresses the Reviewer's concern.

---Point 6:

There are issues of clarity in Figure 2. For instance, the normalization procedure in Figure 2b is inadequately described, and the meaning of the blue spectrum is ambiguous. Figure 2g is difficult to interpret due to the small, unreadable numeric labels and should be redrawn for clarity.

---Reply: We appreciate the Reviewer's constructive feedback. Figure 2 presents simulated data for individual metaatom structures with fixed periodic boundaries ($P_x \times P_y = 125 \mu\text{m} \times 40 \mu\text{m}$), where each spectrum corresponds to an isolated unit cell response. This explains the discrete resonance peaks observed in the original Figure 2b. For clarity, each spectrum in the original Figure 2b was

normalized to obtain transmittance spectra $T(f) = P_{\text{trans}}(f) / P_{\text{inc}}(f)$ ($P_{\text{trans}}(f)$ and $P_{\text{inc}}(f)$ are the transmitted and incident signals with the metasurface), ensuring that the relative resonance features are accurately represented. The blue curve in the original Figure 2b serves as an illustrative example of any single spectrum's characteristic response, highlighting the resonant behavior of a typical metaatom within the gradient metasurface. We have added the normalization procedure in the Methods (the 1st paragraph on page 14).

In addition, we sincerely apologize for the readability issues in the original Figure 2g caused by the small and unclear numeric labels. In the revised manuscript, we have redrawn this Figure R18 (as shown in Figure 2) with enlarged and clearly legible labels to enhance interpretability.

Figure R18. Design of broadband spectral-based plasmonic gradient metasurface.

---Point 7:

There is an inconsistency between the spectra shown in Figure 2f (left) and Figure S2 (top), despite the meta-atoms having identical geometries. This discrepancy needs to be addressed and explained.

---Reply: We thank the Reviewer for bringing this to our attention. We sincerely apologize for causing this confusion. Regarding the apparent inconsistency between the spectra in original Figure 2f (left) and Figure S2 (top), we would like to clarify that the abrupt transmission dip in original Figure 2f (left) arises from the periodic scattering of the metasurface lattice (*Advanced Optical Materials* 2020, **8**(6): 1901572. <https://doi.org/10.1002/adom.201901572>, *Optical Materials Express* 2019, **9**(3). <https://doi.org/10.1364/ome.9.000944>), which is not the primary focus of our study. Both Figures aim to illustrate the physical principles of broadband enhancement, particularly the relationship between the generation of QBICs and the number/type of defects introduced into the

structure.

To better align with the message of the manuscript and minimize ambiguity, we have revised the Figure 2f (as shown in Figure R19) to ensure consistency and improve readability. The updated Figures more clearly emphasize the critical role of defect engineering in achieving continuous spectral coverage through gradient metasurfaces, while de-emphasizing secondary effects like lattice scattering.

Figure R19. Schematic illustration of continuous resonant metasurface whose unit cell is made of two subgroups of gold microbars. The lengths of all microbars are intentionally perturbed to create multi-QBICs. The right part of each schematic is the corresponding SEM image. The corresponding transmission spectrum of the metasurfaces in upper panel. The insets are the mechanism of the interfering resonances for a resonating system.

--Point 8:

Figure 4e is also ambiguous. It is unclear what is being presented and what constitutes the baseline measurement. Additional explanation is required to interpret this figure meaningfully.

--Reply: We thank the Reviewer for pointing this out. The purpose of the original Figure 4e is to illustrate the method used to extract the sample fingerprint shown in the original Figure 4f. In Figure 4e, the solid curve represents the transmission spectrum of the device after loading a mixture of 18 μg Gln, 28 μg GABA, and 18 μg L-Glu. The dashed curve indicates the calibrated baseline, which was obtained using an asymmetric least squares smoothing (AsLSS) algorithm fitting algorithm (*Nature Communications* 2023, 14(1). <https://doi.org/10.1038/s41467-023-43127-z>). Preprocessing of raw spectral measurements initiated with baseline correction to mitigate non-analytical signal components arising from systematic artifacts (e.g., detector noise, Rayleigh scattering, fluorescence background). The adaptive polynomial fitting algorithm implemented in this work operates under the hypothesis that baseline drift can be approximated by low-order polynomial functions (2nd - 4th degree). Mathematically, this is expressed as: $y_{\text{baseline}} = a_0 + a_1x + a_2x^2 + \dots + a_nx^n$. The polynomial coefficients are solved by least squares, and finally the fitted polynomial curves are subtracted from the original spectra: $y_{\text{corrected}} = y_{\text{baseline}} - y_{\text{raw}}$. The area under each spectral peak was quantitatively integrated, revealing that the resonance frequency of the metasurface precisely overlaps with the characteristic absorption peaks of the analyte. Variations in the mass of mixed substances placed at the focal point of the incident THz beam result in systematic and pronounced changes in the transmission spectra. The intensity and depth of each spectral fingerprint scale with the mass of the

analyte: greater quantities of the substance enhance the interaction with the hypersurface, thereby amplifying the spectral response. By subtracting the baseline from the measured spectrum, we can extract the absorption features of the sample, as shown in Figure 4k. To avoid confusion, we have added an explanation of the baseline extraction method and relevant references (*Nature Communications* 2023, 14(1). <https://doi.org/10.1038/s41467-023-43127-z>) in the revised manuscript.

We have now added the details regarding data preprocessing “The dashed curve indicates the calibrated baseline, which was obtained using an asymmetric least square smoothing (AsLSS) fitting algorithm.” in the revised manuscript (the 1st paragraph on page 11).

---Point 9:

The claim made in the manuscript that “It is evident that this set of experiments validates... the THz range” is not adequately supported by the data presented in Figure 4g. In particular, the results do not convincingly demonstrate real-time biomolecule differentiation, since the concentrations of the species are known beforehand and used to interpret the data.

---Reply: We sincerely appreciate the Reviewer’s critical feedback regarding the validation of real-time biomolecule differentiation in the THz range. To address this concern, we conducted additional experiments involving real-time dynamic detection and analysis of the analyte’s absorption fingerprints, along with comprehensive data processing. The results provide robust evidence supporting our assertion.

The Figure R20a demonstrates a dynamic monitoring system integrating microfluidic technology, where time-domain signals were collected every 3 seconds to track electric field intensity changes. These changes precisely correlate with variations in analyte mass (0 μg to 16 μg), as evidenced by the repeatable and quantifiable signal shifts. To rigorously validate the system’s real-time detection capability, we introduced a “blind” test (gray ribbon in Figure R20a), where Researcher A controlled 12 μg of analyte in the microfluid chamber, and Researcher B, without prior knowledge of the analyte mass, performed the measurements. From the experimental data, Researcher B then derive the expected analyte mass, which matches the designed value of 12 μg . This test confirms the system’s ability to successfully differentiate biomolecule concentrations without prior knowledge.

Furthermore, the transmission spectra (Figure R20c), derived from Fast Fourier Transform (FFT) of the time-domain signals, exhibit the specificity of L-Glu characterized by different absorption intensities at 1.22 THz, reinforcing the specificity of our method. Furthermore, the residual analysis validates the system’s reliability, showing minimal deviations from the zero-reference line and a high signal-to-noise ratio ($\text{SNR} \approx 28 \text{ dB}$) throughout the 612-second measurement.

Figure R20. (a) Time-domain signals of electric field amplitude collected at 3-second intervals, demonstrating dynamic changes in response to varying L-Glu masses (0 μg , 4 μg , 6 μg , 14 μg , 16 μg , 12 μg , and 0 μg). The gray band is a prediction of the analyte mass based on the electric field amplitude in the case of unknown analyte mass. (b) The average electric field intensity (the central solid red line) and its stability over 612 seconds of continuous monitoring. The amplitude fluctuations ($\Delta E_{\text{max-min}}$) reflect the sensitivity of the system to analyte mass variations. The red shaded area is a standard deviation. (c) Transmission spectra obtained by FFT of the time-domain signals showed the specificity of L-Glu characterized by different absorption intensities at 1.22 THz. (d) The residual analysis confirms minimal deviation from the zero-reference line, indicating high signal-to-noise ratio (SNR) and system stability for long-term measurements.

These new experimental data are now included in the modified Figures 4d - 4g (the caption of the Figure 4 in page 9). The main text has also been updated accordingly (in the 2nd paragraph of page 10).

---Point 10:

The supplementary figure S3 presents simulated spectra for two “super meta-atoms,” yet this concept is not introduced or discussed in the main manuscript. Instead, the text refers to principal component loadings, which appear to correspond to Figure S4. This misalignment needs correction for consistency and clarity.

---Reply: We thank the Reviewers for these points. We sincerely apologize for any confusion caused by the misalignment between the supplementary figures and the main text. We appreciate the Reviewer’s careful attention to this detail. We would like to emphasize that Figure S3 presents the experimental results of two super-metatoms, which serve as key components of the gradient metasurface design illustrated in Figure 2a of the main manuscript. These super-metatoms were

specifically engineered to validate the broadband enhancement capabilities of our structure. The simulated spectra in Figure S3 further illustrate how varying the geometric parameters of these super-metaatoms enables tailored spectral coverage, aligning with the broader goals of the study.

As presented in Figure S4, we evaluated the score weights of each spectrum in a two-dimensional space using PCA. To ensure that the spectral data adequately capture and represent the biological characteristics of each sample, we performed PCA on the entire spectral dataset. This analysis enabled us to cluster the samples and visualize the correlations and distinctions among them, thereby confirming that the spectral features effectively reflect the intrinsic biological information. Corresponding revisions have been made and clearly labeled in the main text (2nd paragraph, 12 page). We have conducted a thorough review of the entire manuscript to ensure consistency between the main text and Supplement Information.

Accordingly, we have added the illustration of the “*super meta-atoms*” in the Figure 1 caption on page 5. In addition, we have conducted a thorough review of the entire manuscript to ensure consistency between the main text and Supplement Information.

--Point 11:

The spatial barcode spectra need to be described in more detail, and the associated color bar should be added to aid interpretation. Furthermore, absorption spectra of the analytes at varying concentrations, without the metasurface, should be provided as a necessary reference for understanding their intrinsic spectral features.

--Reply: We thank Reviewer for these constructive comments. The spatial barcode spectra were obtained by measuring the absorption spectra of analyte powders pressed into pellets, ensuring a clear representation of their intrinsic spectral features. To further enhance the interpretability of the data, we have included corresponding color bars, as shown in the Figures R21, which provides a visual guide to the spectral intensity variations.

Figure R21. Absorption barcodes of experimentally measured spectra of three species pure/mixed biomolecules. Color bar was added to the Figure 4c.

Additionally, we sincerely appreciate the Reviewer's constructive suggestion regarding the need for

reference absorption spectra without the metasurface. As shown in the Figure R22, we have systematically measured the intrinsic transmission spectra (main panel) and corresponding time-domain signals (inset) for: i) pure solvent, ii) GABA (28 μg), iii) GABA (28 μg) with glutamate (3 μg) and glutamine (5 μg), and iv) GABA (28 μg) with glutamate (6 μg) and glutamine (10 μg). These reference measurements clearly reveal the characteristic absorption features of each analyte in the 1.2-2.0 THz range, with the time-domain signals (0.2 to -0.2 V) demonstrating excellent stability over 38-44 ps. Figures 4h and 4i present the transmission spectra without and with the metasurface for analytes, respectively. A direct comparison demonstrates significant signal enhancement at characteristic absorption positions when using the metasurface. For example, the THz-TDS spectrum of the mixture in solution (Figure R21a) shows broad, weak absorption features (e.g., 1.53 THz for GABA). Next, we evaluate the enhancement factor (EF) of the metasensor.

The EF of the enhanced THz absorption spectroscopy (ETHzAS) based on gradient metasurface metasensor is quantitatively defined to characterize its performance in amplifying molecular absorption signals. The EF is calculated using the following relationship (*Advanced Science (Weinh)* 2021, 8(20): e2101879. <https://doi.org/10.1002/advs.202101879>):

$$EF = \frac{I_{ETHzAS} - I_{ref}}{I_{ref}} \times \frac{N_{ref}}{N_{ETHzAS}} \quad (\text{R4})$$

Among them, I_{ETHzAS} represents the absorption intensity measured with the metasensor, while I_{ref} denotes the reference absorption intensity obtained without the metasurface. The term N_{ref} corresponds to the number of molecules contributing to absorption in the reference case (e.g., molecules in the microfluidic solution), and N_{ETHzAS} refers to the number of molecules within the metasurface's localized field-enhanced region.

To compute the EF, we first extract the absorption intensities at the target frequency (1.54 THz for GABA). For instance, if the reference absorption I_{ref} is 1.35% (Figure R22b) and the sensor-enhanced absorption I_{ETHzAS} is 1.72% (Figure 4k), the signal enhancement $I_{ETHzAS} - I_{ref}$ equals 0.37%. To calculate the molecular ratio, we compare the effective interaction volumes under identical illumination conditions (same beam spot size), focusing exclusively on thickness differences. The reference volume V_{ref} spans the full microfluidic channel thickness (1 mm = 1000 μm), while the enhanced volume V_{ETHzAS} is now defined by the metasurface's interaction depth of $d = 3.5 \mu\text{m}$ (Figure R22c). So, the EF is: $EF \approx 80$.

These results provide conclusive evidence that our sensor not only preserves the intrinsic spectral features of the analytes, but also significantly enhances their detection sensitivity through metasurface integration. The combination of both datasets offers a comprehensive understanding of both the fundamental molecular properties and the enhanced sensing capabilities of our platform.

Figure R22. (a) Reference transmission spectra and time-domain signals of analytes without metasurface enhancement. The main panel shows the intrinsic THz transmission spectra of: i) pure solvent, ii) GABA (28 μg), iii) GABA (28 μg) with L-Glu (3 μg) and Gln (5 μg), and iv) GABA (28 μg) with L-Glu (6 μg) and Gln (10 μg) in the 1.2–2.0 THz range. The inset displays the corresponding time-domain electric field signals. Characteristic absorption features of each analyte are clearly resolved, providing essential baseline data for evaluating metasurface-enhanced detection (Figure 4j). (b) Absorption spectrum calculated from the transmission spectra in (a). (c) The electric field distribution and intensity in both the xy and xz planes of the structural unit ($L_1 = 48 \mu\text{m}$) were monitored separately. The black solid line represents the 1/e contour of the maximum enhanced electric field. $d = 3.5 \mu\text{m}$ is the height from the position of the 1/e electric field intensity to the metasurface surface.

We believe these additions significantly improve the clarity and robustness of our results, addressing the Reviewer's concerns and providing a more comprehensive understanding of the spectral data. The updated Figures R21 has been added in the revised manuscript as Figure 4c on page 9. In addition, we added the Figure R22a as the Figure 4h in the revised manuscript (in the 2nd paragraph

of page 11). What's more, we added an evaluation of the enhanced THz absorption spectroscopy (ETHzAS) enhancement factor (EF) for gradient metasurface element-based sensors in the revised manuscript (in the 2nd on the page 11) and Figures R22b and R22c to the Supplement Information as Figures S6a and S6a.

---Point 12:

Some terminologies used in the manuscript also require clarification. The meaning of the term “tuning speed” is vague and should be defined precisely. Additionally, the type of solvent used for dissolving the biomolecules should be explicitly stated.

---Reply: We thank Reviewer for this constructive comment. We agree with the Reviewer that the term “tuning speed” is poorly worded and vague. We have replaced the phrase “tuning speed” with “The gradient variation rate of the metaatom dimensions” throughout the manuscript.

And regarding the type of solvent, we have added the following details in 2nd of Page 10 in the revised manuscript. The solvent used in this study was edible oil, which served as a dispersion medium for the biomolecules (L-Glu, GABA, and Gln) and facilitated their controlled delivery onto the plasmonic gradient metasurface. As shown in Figure 4c of the manuscript, the oil-coated metasurface exhibited a broad transmission spectrum (Figure 4b, orange line) with reduced intensity due to the oil's inherent THz absorption, but no solvent-specific peaks interfered with the target biomolecular fingerprints. This choice of solvent was critical for two reasons: i) to disperse the biomolecules uniformly for consistent THz absorption measurements, ii) its chemical inertness ensured minimal interaction with the analytes or metasurface, preserving the integrity of the vibrational fingerprints. The oil's homogeneous dispersion of biomolecules further allowed for reproducible detection of concentration-dependent spectral modulations (Figures 4f and 4i), validating its suitability for label-free, in-situ THz biosensing.

We trust that we have adequately addressed all the issues raised by the Reviewers. Many thanks for your kind help and effort on our article. We hope our paper can now be accepted for publication. Please feel free to contact and advise us.

Best Regards,

Din Ping Tsai,

Dear Reviewers,

We sincerely thank the reviewers for your positive evaluation and kind decision on our manuscript submitted to *Nature Communications* (NCOMMS-25-26168A). We have thoroughly revised the manuscript according to the suggested guidelines, provided detailed point-by-point responses in blue, and highlighted all changes in the text to facilitate the reviewer #3's evaluation.

The following is a point-by-point response to the Reviewers' comments

REVIEWERS' COMMENTS

To Referee #1:

The authors have provided thorough and satisfactory responses to my comments, and the revisions appropriately address the issues I raised. Accordingly, I regard the outcome of my review comments as positive.

---Reply: We are grateful to the reviewers for their recognition of our work.

To Referee #2:

The authors have carefully addressed all of my previous concerns, as well as those raised by the other reviewers. The revised manuscript shows significant improvements in clarity, scientific accuracy, and overall presentation. The responses are detailed and convincing, and the changes to the manuscript adequately reflect the reviewers' feedback.

In my opinion, the manuscript is now suitable for publication in its current form.

---Reply: We appreciate the reviewer for the encouraging feedbacks.

To Referee #3:

The authors addressed in detail the previous comments however there are still some issues to be solved before accepting the manuscript for publication.

---Reply: We thank the reviewer for their positive comments on our previous response and revision, and for their constructive insights which help us to further improve quality of the manuscript. In the following and the revised manuscript, we have modified the article in accordance with the full sets of the reviewer's comments.

1) In the reply to point 2, the authors stated that they clearly define the nature of the plasmonic resonance in the revised manuscript. However, it is not clear in which section they add this description.

---Reply: We thank the Reviewer for pointing this out. We deeply apologize for the unclear labeling of the corresponding paragraphs in our initial revised manuscript. Therefore, we have explicitly added markers in our second revision, which are located in the 1st paragraph on page 2, the 3rd paragraph on page 3, the 2nd paragraph on page 4, the caption of Figure 1, the 1st paragraph on page 5,

the caption of Figure 2, 2nd and 3rd paragraphs on page 7, the 1st paragraph on page 10, the 3rd paragraph on page 11, and the 3rd paragraph on page 13 in the original version.

2) *I suggest to report part of the explanation about the Q-factor value (Reply to point 3) in the revised manuscript.*

---Reply: We sincerely thank the Reviewer for the valuable suggestion. we have incorporated a relevant explanation into the 1st paragraph of Page 7 in the revised manuscript. The added content reads: “In this system, the relatively low Q-factor is primarily attributed to scattering and material losses resulting from the inevitable surface roughness of the metasurface and its finite area, which collectively limit the resonance linewidth.”

3) *Regarding previous point 4, there are still some differences between Figure 2c and Figure 4b. In particular, in the latter case the transmission at about 2 THz is very low compared to Figure 2c. A simulation of the super-metaatom will help to clarify this point.*

---Reply: We are grateful for the Reviewer’s insightful comment. Some discrepancies between them are to be expected due to both experimental limitation and simulation differences. Firstly, they are subject to experimental constraints. This is the primary reason why the experimental result in Figure 4c cannot fully resolve individual modes—a phenomenon we have consistently observed and clarified in our previous studies on multi-band metamaterial structures, as shown in Figure R1. As can be seen from Figures R1(c) and (d), different modes can be distinguished with a spacing of 291.8 GHz. Meanwhile, the differences become more pronounced in the high-frequency region. In contrast, during simulation, our design employs 19 unit-cells to span 0.9 THz, yielding a resonance density of 21 modes/THz (1 mode per 47 GHz). The spectral resolution in simulation can be further improved by increasing the number of frequency sampling points. In addition, in the simulation, the metal was modeled as a perfect electric conductor (PEC), as explained in our response to Point 5 in the 1st round of review. The simulation results for several gradient structures are provided in Panel 4 of Figure S2 (Supporting Information). Experimentally, however, unavoidable scattering loss due to surface roughness and the finite size of the metasurface (as shown in Figure S2) leads to a reduction in the total Q-factor. This can be quantitatively described by the relation $Q^{-1} = Q_r^{-1} + Q_{nr}^{-1}$, where Q , Q_r , and Q_{nr} represent the total, radiative, and non-radiative Q-factors, respectively. Scattering and material losses therefore broaden the resonance linewidth and reduce transmission intensity, particularly higher frequency (around 2 THz), accounting for the observed differences between simulation and experiment.

Figure R1 (a) Schematic of the multi-band QBICs metasurface. (b) Detailed parameters of different microbars. (c) and (d) Simulated and experimental reflectance spectra of the unit cell shown in (a).

4) The quality of Figure 4 needs to be improved because the text and line in some panel, eg. H, are not very clear.

---Reply: We greatly appreciate the Reviewer's suggestion. We sincerely apologize for the lack of clarity in some panels of Figure 4, particularly panel H, which caused inconvenience to the readers. We have carefully revised Figure 4H to improve the resolution and legibility of both text and lines. In addition, we have thoroughly reviewed all figures throughout the manuscript to ensure they meet the high standards of the journal and enhance the overall readability of the article.

Brief summary of the main findings of the paper

This research creates a terahertz plasmonic gradient metasurface for miniaturized spectroscopy. It enables continuous, broad spectral sensing without scanning, allowing real-time, label-free identification of multiple biomolecules.

Thank you for your consideration!

Sincerely,

Prof. Din Ping Tsai